# EMBO *reports*

# Report

# Muscle-derived exophers promote reproductive fitness

Michał Turek[1,2,3,*] iD, Katarzyna Banasiak[4] iD, Małgorzata Piechota[4,†] iD, Nilesh Shanmugam[4,†] iD, Matylda Macias[5] iD, Małgorzata Alicja Śliwińska[6] iD, Marta Niklewicz[4] iD, Konrad Kowalski[4] iD, Natalia Nowak[6] iD, Agnieszka Chacinska[1,2,7] iD & Wojciech Pokrzywa[4,**] iD

## Abstract

Organismal functionality and reproduction depend on metabolic rewiring and balanced energy resources. However, the crosstalk between organismal homeostasis and fecundity and the associated paracrine signaling mechanisms are still poorly understood. Using *Caenorhabditis elegans*, we discovered that large extracellular vesicles (known as exophers) previously found to remove damaged subcellular elements in neurons and cardiomyocytes are released by body wall muscles (BWM) to support embryonic growth. Exopher formation (exopheresis) by BWM is sex-specific and a non-cell autonomous process regulated by developing embryos in the uterus. Embryo-derived factors induce the production of exophers that transport yolk proteins produced in the BWM and ultimately deliver them to newly formed oocytes. Consequently, offspring of mothers with a high number of muscle-derived exophers grew faster. We propose that the primary role of muscular exopheresis is to stimulate reproductive capacity, thereby influencing the adaptation of worm populations to the current environmental conditions.

**Keywords** exophers; intertissue signaling; muscle; vesicular transport; vitellogenin
**Subject Categories** Development; Membranes & Trafficking; Musculoskeletal System
See also: **EJ Cram** (August 2021)

## Introduction

The proper cellular function relies on removing unwanted contents by proteolysis or degradation mainly via the ubiquitin-proteasome system (UPS) and autophagy (Dikic, 2017). Recently, a complementary mechanism was described in *Caenorhabditis elegans*. Under proteotoxic stress, neurons remove protein aggregates and damaged mitochondria via large membrane-bound vesicles called exophers. This process is beneficial for neuron functionality as cells that generate exophers perform better than those devoid of this mechanism (Melentijevic *et al*, 2017). However, other cells, in addition to neurons, can remove cellular content via exophers. Murine cardiomyocytes can eject subcellular content along with mitochondria via exopher-like structures that are ultimately taken up and eliminated by cardiac macrophages (Nicolás-Ávila *et al*, 2020). This extrusion phenomenon constitutes a significant but still poorly explored metabolic waste management pathway. Here, we report that *C. elegans* body wall muscle (BWM) cells can produce exophers in a sex-specific manner. Furthermore, the generation of muscular exophers depends on developing embryos in the uterus and positively correlates with the number of retained embryos. Finally, we show that exophers serve as transporters for muscle-generated yolk proteins, which support offspring development. We identified a new role for exopheresis that extends beyond the removal of proteotoxic cellular components and has a transgenerational effect on the animal population.

## Results and Discussion

### *Caenorhabditis elegans* muscles remove cellular content via exophers

Using worms expressing fluorescent reporters in the BWM cells, we identified muscle-derived vesicles resembling exophers. These reporters allow simultaneous tracking of mitochondria, which can be extruded in neuronal and cardiac exophers (Melentijevic *et al*, 2017; Nicolas-Avila *et al*, 2020) and proteasomes, a key component

1  ReMedy International Research Agenda Unit, University of Warsaw, Warsaw, Poland
2  Laboratory of Mitochondrial Biogenesis, Centre of New Technologies, University of Warsaw, Warsaw, Poland
3  Laboratory of Animal Molecular Physiology, Institute of Biochemistry and Biophysics, Polish Academy of Sciences, Warsaw, Poland
4  Laboratory of Protein Metabolism, International Institute of Molecular and Cell Biology, Warsaw, Poland
5  Core Facility, International Institute of Molecular and Cell Biology in Warsaw, Warsaw, Poland
6  Laboratory of Imaging Tissue Structure and Function, Nencki Institute of Experimental Biology Polish Academy of Sciences, Warsaw, Poland
7  IMol Polish Academy of Sciences, Warsaw, Poland
   *Corresponding author. Tel: +48 22 592 20 24; E-mail: m.turek@ibb.waw.pl
   **Corresponding author. Tel: +48 22 597 07 43; E-mail: wpokrzywa@iimcb.gov.pl
   †These authors contributed equally to this work

of the proteostasis network. Like *C. elegans* neurons, muscles generate a small fraction (approximately 12%) of the vesicles containing the mitochondria, but all of those we examined were filled with proteasomal subunits that are components of the 19S and 20S proteasome (Figs 1A and EV1A). The size of the described vesicles extruded by BWM cells also corresponds to neural exophers

(Melentijevic *et al*, 2017). However, those of muscular origin are generally more abundant and prominent (with diameters ranging from 2 to 15 μm; Fig 1B, Appendix Table S1). As with neural exopheresis, muscle exopheresis is not limited to a particular reporter or transgenic line (Figs 1A, B and D, 2C and 4H, Appendix Table S2). Moreover, our electron microscopy (EM)

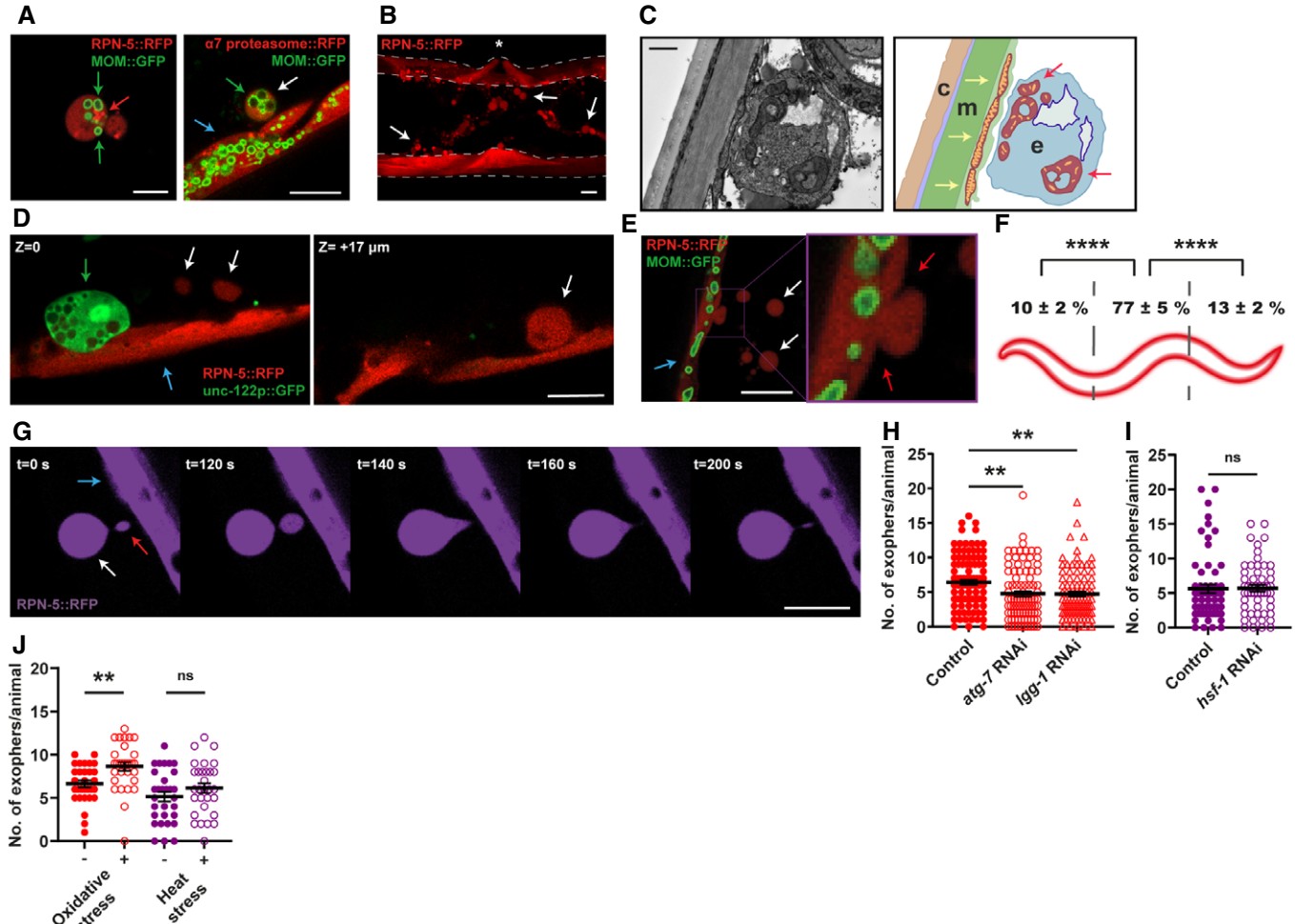

**Figure 1. *Caenorhabditis elegans* muscles expel cellular content via exophers.**

A  Muscular exophers contain organelles and large protein complexes. Arrows: white—exopher, blue—muscle cell, green—mitochondria, red—proteasome foci. MOM—mitochondrial outer membrane.

B  BWM actively releases significant amounts of exophers. The image shows the middle part of the worm's body with muscles marked with dashed lines. Arrows indicate representative exophers, and the asterisk indicates the position of the vulva.

C  The ultrastructure of the muscular exopher and its schematic view. Arrows: red—morphologically changed mitochondria inside the exopher, yellow—normal, elongated mitochondria inside the muscle cell. c—cuticle, m—muscle, e—exopher.

D  Comparison between muscular exopher and coelomocyte. Arrows: white—exopher, blue—muscle, green—coelomocyte.

E  Exophers are formed via a pinching-off mechanism. Arrows: white—exopher, blue—muscle cell, red—distorted muscle cell membrane during exopher formation.

F  Production of muscular exophers is not evenly distributed across all muscle cells. The highest number of exophers is produced by the muscles adjacent to the vulva. *n* = 46; *N* = 3.

G  Exophers may remain connected to the sending BWM cells via thin elastic tubes that allow further transfer of cellular material. Arrows: white—exopher, blue—muscle cell, red—cellular material transferred to exopher via elastic tube.

H  Knockdown of two autophagy genes, *atg-7* and *lgg-1* significantly reduces the number of generated exophers. *n* = 91–103; *N* = 3.

I  Proteostasis disruption by *hsf-1* knockdown does not increase exopher production. *n* = 60 and 55; *N* = 2.

J  Challenging proteostasis via oxidative stress but not heat stress increases exophers production. *n* = 30; *N* = 3.

Data information: Scale bars are 10 μm (A, B, D, E, G) and 1 μm (C). Data are shown as mean ± SEM; *n* represents the number of worms; *N* represents the number of experimental repeats combined into a single value; ns—not significant, **P < 0.01, ****P < 0.0001; (F, H, I, J) Mann–Whitney test.

studies visualized membrane-bound exophers adjacent to the BWM. Vesicles analyzed by EM show similar compactness, with the characteristic presence of mitochondria (Figs 1C and EV1C and D). Most of them showed marked morphology changes, characterized by increased area and disturbed cristae organization as in cardiac exophers. However, we noted the presence of apparently intact mitochondria in BWM-derived structures, suggesting that they are not only used to transport defective organelles. To confirm that the identified vesicles are the counterpart of exophers, we tracked their formation in response to RNAi knockdown of genes that regulate neuronal exopheresis (Melentijevic *et al*, 2017). Indeed, the appearance of muscle-derived exophers was also affected by depletion of NAPH-cytochrome P450 reductase EMB-8 and actin-binding protein POD-1 (Fig EV1B). Finally, we have also excluded the possibility that observed vesicles are part of coelomocytes (Fig 1D).

Exophers are formed in the muscle cell and expelled outside via a pinching-off mechanism (Fig 1E), and the majority are generated by adult hermaphrodite mid-body muscles (Fig 1F). Some exophers that bud off remain connected with the extruding BWM via a thin but elastic tube that permits the continued transfer of a large amount of cellular material into the extruded vesicle (Fig 1G, Movie EV1), similar to neuronal exophers (Melentijevic *et al*, 2017). We next examined whether the autophagy machinery was involved in the generation of exophers. Indeed, the number of exophers decreased when autophagy components were knocked down (Fig 1H), indicating crosstalk between autophagy and exopheresis. Proteostasis impairment significantly increases neuronal exopher output (Melentijevic *et al*, 2017; preprint: Hualin *et al*, 2019). By contrast, the number of muscular exophers did not change in response to depletion of the central proteostasis transcription factor HSF-1 (via *hsf-1* RNAi) or heat stress, and the number of exophers increased slightly under conditions of oxidative stress (Fig 1I and J). These observations suggest that proteostasis regulation might not be the core function of muscle exopheresis.

## Muscular exopheresis is a sex-specific process regulated in a non-cell autonomous manner

Next, we assessed the number of exophers at different time points of the *C. elegans* hermaphrodite life cycle. Reminiscent of neuronal exophers, muscular exophers are not produced during the larval stages, and their maximum level is reached around the second and third days of hermaphrodite adulthood (Fig 2A). Because this time point coincides with the worm's maximum reproductive rate, we wondered if reproduction could influence exopher formation. To examine this possibility, we followed exopheresis in males. For the first 3 days of adulthood, males did not produce any exophers (Fig 2A). This finding suggests that germ cell maturation in the reproductive system of hermaphrodite worms, the process of oocyte fertilization, or embryonic development might regulate muscle exopheresis.

To test these hypotheses, we took advantage of a thermosensitive *fem-1* mutant strain that does not produce viable sperm at the restrictive temperature of 25°C. At the permissive temperature of 15°C, some animals can reproduce like wild-type hermaphrodites, whereas the rest of the population is sterile (Nelson *et al*, 1978). The offspring-producing *fem-1* mutant animals grown at 15°C

generated a high number of muscular exophers. By contrast, animals raised at either 15 or 25°C that were unable to fertilize oocytes did not activate muscular exopheresis (Fig 2B and C), indicating that neither the female gonad nor the temperature itself was sufficient to trigger exopher release. However, re-establishing fertility in *fem-1* mutants by mating them to *him*-5 mutant males restored exopher production at the restrictive temperature (Fig 2B). Moreover, hermaphrodites sterilized via fluorodeoxyuridine (FUdR) treatment (Hosono, 1978) extruded no exophers or only a few per animal (Fig 2D), suggesting that the occurrence of developing embryos could indeed stimulate muscular exopheresis. We also found that hermaphrodites treated with FUdR often contained structures in their BWM that appeared like segregated exopher cargo (Fig 2E, middle and right panels). Interestingly, we detected similar objects in males (Fig 2E, left panel) that, as in sterile hermaphrodites, are not excreted by the BWM. The above results suggest that the occurrence of developing embryos could be required to induce muscular exopheresis. Consistently, we observed a positive correlation between the number of exophers released and the number of embryos present in the uterus (Fig 3A). To further explore this link, we depleted the mRNA of genes responsible for various processes associated with egg-laying. RNAi depletion of the G-protein signaling gene *goa-1* leads to hyperactive egg-laying behavior, resulting in the presence of fewer early-stage embryos within the uterus (Bany *et al*, 2003). *goa-1* knockdown caused a significant drop in the level of exopher release. By contrast, the egg-laying defects induced by *egl-1* and *egl-4* RNAi, which lead to embryo retention in the uterus (Hirose *et al*, 2003), increased exopher formation by muscle cells (Fig 3B). In the absence of food, worms halt egg-laying and retain embryos in the uterus (Daniels *et al*, 2000; Dong *et al*, 2000); therefore, we anticipated that worms would generate more exophers when experiencing a food shortage. Indeed, the accumulation of developing embryos in the uterus caused by the transfer of adult worms to food-free plates resulted in a significant increase in muscle exopher secretion in contrast to L4 larvae grown under conditions of food shortage (Fig 3C).

Next, we tested whether worm embryos could directly induce exopher production. We incubated young adult worms in medium conditioned with developing embryos or in the extract obtained from wild-type embryos or L2/L3 larvae (Fig 3D). Intriguingly, exopheresis was significantly increased in worms exposed to the embryo secretome or material derived from their lysis (Figs 3E and EV2A). This effect was specific to hermaphrodites, as incubation of males with embryo extract did not induce exopher formation (Fig EV2A). Moreover, increased permeability of the eggshell via *emb-27* RNAi knockdown (Sato *et al*, 2008) robustly elevated muscular exopheresis (Fig 3F). This result suggests that molecules that diffuse from embryos *in utero* are responsible for exopheresis induction in hermaphrodites. We next examined whether mechanical stretching of the uterine muscles or body wall by accumulated embryos could contribute to increased exopher production. To this end, we knocked down *egl-1* in animals that were cultured on FUdR-supplemented plates, which resulted in the retention of dead embryos in the uterus and its mechanical stretching. However, these worms did not produce exophers (Fig EV2B and C), indicating that it is not mechanical impact but the presence of developing embryos in the uterus that is responsible for the induction of exopheresis.

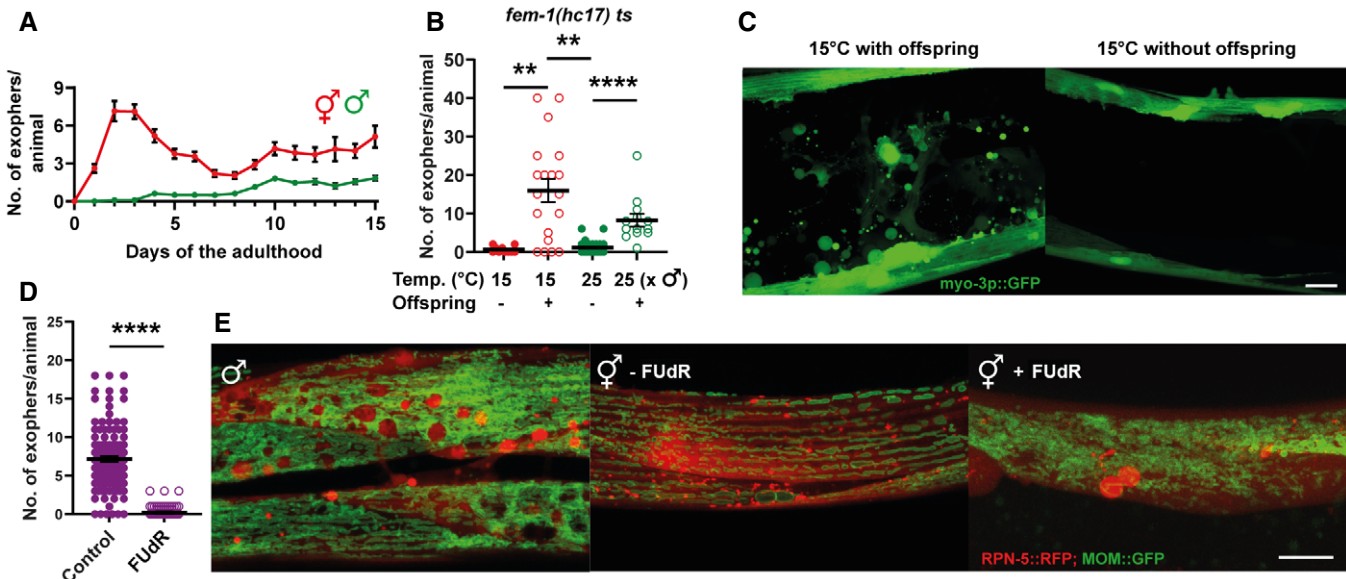

**Figure 2. Exopher formation is sex-specific and fertility-dependent.**

A   The highest number of exophers is produced during the hermaphrodite reproductive period and in aging animals. Males do not produce exophers during the first days of adulthood and begin to generate a small number of exophers later in life. Starting n = 90 hermaphrodites and 150 males; N = 3.
B   Feminized hermaphrodites of a thermosensitive *fem-1* mutant strain do not produce exophers regardless of growth temperature. This phenotype can be partially rescued by mating *fem-1* mutants with males. n = 10–26; N = 2.
C   Representative images of the middle part of the worm body in panel B.
D   Hermaphrodites sterilized via FUdR treatment produce no exophers or only a few per animal. n = 118 and 112 animals; N = 3.
E   Males and sterile hermaphrodites (via FUdR treatment) show the formation of spherical structures in the BWM that resemble mature exophers. MOM—mitochondrial outer membrane.

Data information: Scale bars are 10 μm. Data are shown as mean ± SEM; n represents the number of worms; N represents the number of experimental repeats combined into a single value; **P < 0.01, ****P < 0.0001; (B, D) Mann–Whitney test.

Finally, we investigated whether disturbed proteostasis specifically in the BWM or a signal associated with the developing embryos in the uterus would be the primary regulator of muscular exopheresis. To this end, we knocked down the myosin-directed chaperone UNC-45, which results in defects in the folding and assembly of myosin thick filaments as well as embryonic defects in cytokinesis and polarity determination (Kachur *et al*, 2004; Gazda *et al*, 2013; Pokrzywa & Hoppe, 2013). We initiated UNC-45 depletion in L4 larvae, which first leads to a disturbance of proteostasis in BWM, and later, in young gravid adults, which inhibits embryonic development. Despite dysfunction of the myosin-chaperone network in muscle cells, which was indicated by complete paralysis, we observed a dramatic inhibition of exopher formation (Fig EV2D). Moreover, depletion of UNC-54, a main client of the UNC-45 chaperone (Barral & Epstein, 1999; Melkani *et al*, 2011), encoding a myosin heavy chain class II (MHC-B) that is expressed in BWM, did not affect exopheresis activity (Fig EV2B). These results point to a predominant role of *in utero* developing embryos in muscular exopheresis induction.

**Exophers transport muscle-synthesized yolk proteins and support offspring development**

The high number of exophers produced by BWM over the *C. elegans* lifespan probably leads to removing a significant

portion of subcellular components by a single tissue. Thus, we hypothesized that this process should have substantial effects on worm muscle functionality and health span. To this end, we selected three types of worms from a synchronized population of gravid adults based on the number of extruded exophers, i.e., few (< 2), many (> 20), and control (6–8) animals (Fig 4A) and analyzed their locomotion. In neurons, exopher production is correlated with improved cell functionality; however, worms with intensified exopheresis did not show enhanced mobility (Figs 4B and EV3). On the contrary, these animals displayed a reduction in exploratory locomotion state (Fig 4C and D). Since exopher production does not seem to have a positive effect on the functionality of the muscles, we next wondered whether, in connection with the role of embryos in exopheresis, this process benefited the offspring. Previous work showed that neuronal exopher content could be transported through the worm body to reach distant scavenger cells (coelomocytes) (Melentijevic *et al*, 2017). However, the fate of muscle exophers may be different from neuronal counterparts, as the exopheresis level does not depend on the apoptotic engulfment pathway proteins CED-1 and CED-6 (Fig EV4A). Muscle-specific transcriptomic analysis revealed the presence of significant levels of vitellogenin mRNAs (i.e., *vit-2*, *-5*, and *-6*) (Blazie *et al*, 2015). Therefore, we hypothesized that the yolk components from BWM are transported through exophers to be used as a source of raw materials for

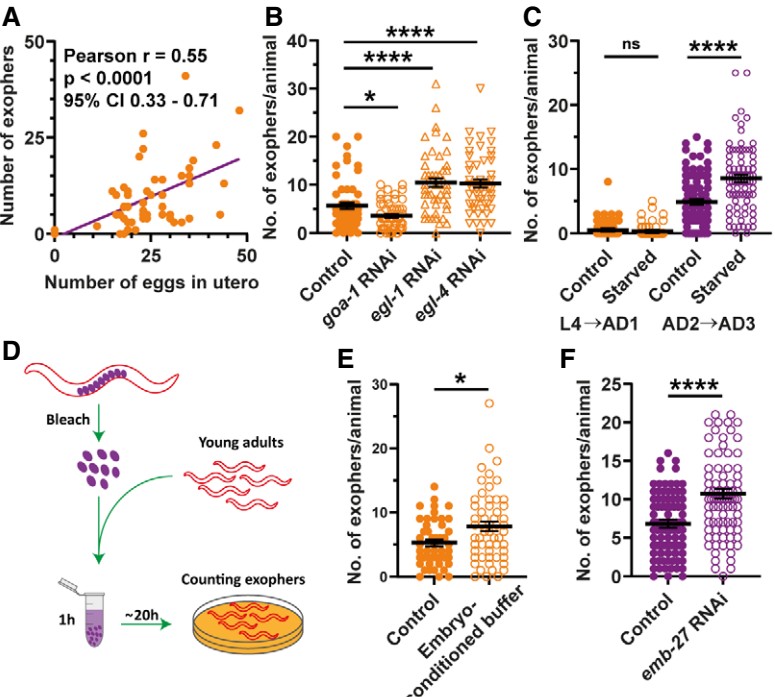

**Figure 3. Muscular exopheresis is a non-cell-autonomous process regulated by *in utero-developing* embryos.**

A   The number of produced exophers positively correlates with the number of *in utero* embryos. The violet line is a linear regression line, and each orange point represents one animal. All animals were 1–3 days old; *n* = 54; *N* = 3.
B   RNAi knockdown of genes regulating the egg-laying rate and their presence in the uterus influences exopher production. *n* = 50–60; *N* = 2.
C   Embryos retention in the uterus caused by starvation during worm's reproductive period increases exopher production. *n* = 81–90; *N* = 3.
D   Schematic representation of the experimental setup for investigating the influence of embryo-conditioned buffer on exopher production.
E   Exposure of young adult worms to embryo-derived substances increases exopher production. *n* = 48 and 56; *N* = 3.
F   Increased eggshell permeability caused by *emb-27* knockdown elevates exopher production. *n* = 103 and 89; *N* = 3.

Data information: Data are shown as mean ± SEM; *n* represents the number of worms; *N* represents number of experimental repeats that were combined into a single value; ns—not significant, *$P < 0.05$, ****$P < 0.0001$; (A) two-tailed Student's *t*-test, (B, C, E, F) Mann–Whitney test.

maturing oocytes. To address this possibility, we used RNAi knockdown of *vit-1* (a vitellogenin-coding gene) to deplete the principal intestinal yolk protein (Perez & Lehner, 2019) in worms. The number of accumulated muscle-released exophers nearly doubled in response to yolk protein depletion (Fig 4E), suggesting a possible compensatory mechanism. Moreover, the intensification of exopheresis in the mother increased the amount of vitellogenin in embryos (Fig 4F). The yolk is transported to proximal oocytes via RME-2 receptor-mediated endocytosis (Grant & Hirsh, 1999; Hall *et al*, 1999); therefore, we tested for RME-2 receptor involvement in the regulation of exopheresis. Depletion of the RME-2 yolk receptor and subsequent inhibition of yolk uptake by oocytes led to a drastic reduction in exopheresis (Fig 4G), which cannot be explained by the sterility of worms, as embryo viability was not entirely compromised even in the *rme-2* deletion mutant (*b1008*; more than 20% of embryos survive) (Grant & Hirsh, 1999). Finally, we could observe that although *ced-1* RNAi did not lead to an increased exopheresis level (Fig EV4A), it caused a slight elevation of VIT-2::GFP in embryos (by approx. 9%; Fig EV4B), which might indicate that a small proportion of exophers transporting cargo may undergo phagocytosis.

Next, we followed the localization of endogenous vitellogenin-2 (VIT-2) fused to GFP (CRISPR/Cas9 knock-in strain) and muscle exopher markers in the hermaphrodite. We detected the presence of endogenous VIT-2::GFP in the BWM of day-2 adult worms, as well as a significant accumulation in many muscle exophers (Figs 4H and EV4C). These results suggest that exophers can mediate the transport of additional portions of muscle-produced vitellogenin, which is ultimately deposited in oocytes from the body cavity (Hall *et al*, 1999). Finally, we followed the growth of the progeny of hermaphrodites exhibiting different levels of exopheresis (Fig 4A). We found that offspring from mothers with a high number of muscle-released exophers grew faster (Fig 4I), consistent with previous reports showing that yolk-rich embryos support post-embryonic survival and larval development (Van Rompay *et al*, 2015; Perez *et al*, 2017; Perez & Lehner, 2019).

Recent reports indicate that the class of the largest extracellular vesicles, known as exophers, are responsible for removing neurotoxic components in neurons (Melentijevic *et al*, 2017) and damaged mitochondria in cardiomyocytes (Nicolas-Avila *et al*, 2020). Here, we show that exophers are not only a storage compartment for cellular waste but that muscular exopheresis in

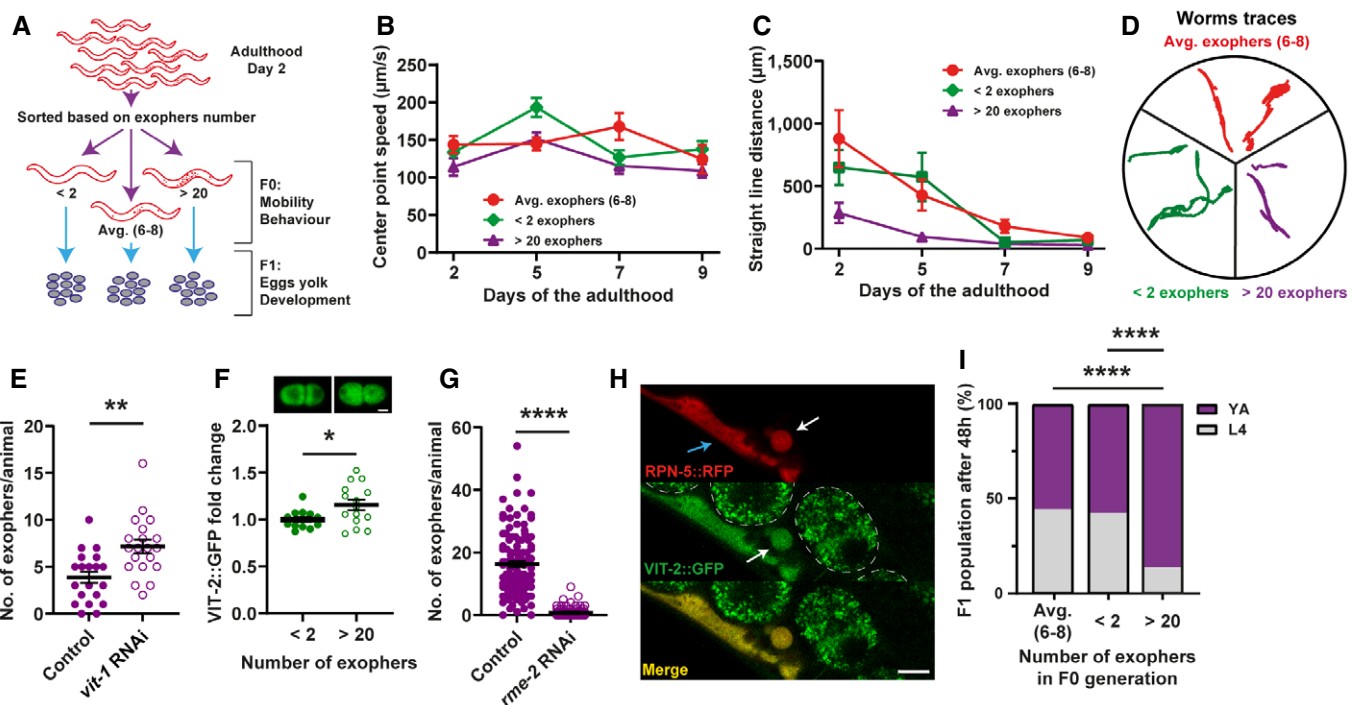

**Figure 4. Muscular exopheresis benefits offspring development.**

A    Schematic representation of the experimental setup for investigating the influence of overactive exopheresis on F0 and F1 worm generation.

B    Exopher production does not improve muscle functionality. $n = 11–38$; $N = 3$.

C, D  Animals with overactive exopheresis have reduced exploratory behavior presented as a reduction in straight-line distance traveled (C) and by representative worms traces on the plate (D). $n = 11–34$; $N = 3$.

E    RNAi knockdown of the egg yolk precursor protein VIT-1 increases the number of muscular exophers. $n = 20$; $N = 2$.

F    Embryos from hermaphrodite mothers that produce a high number of exophers contain more egg yolk precursor protein VIT-2. Representative images of embryos with endogenous VIT-2::GFP levels from mothers with different exopheresis activity are shown above the graph. $n = 15$ and $14$, $N = 5$ and $6$. Scale bar is 10 μm.

G    RNAi knockdown of RME-2 yolk receptor abolish exopher production. $n = 110–123$; $N = 3$.

H    Muscle-produced VIT-2 is released from muscles via exophers. The image shows the mid-body of worms expressing the proteasome subunit RPN-5 tagged with RFP in BWM and VIT-2::GFP endogenous expression. Arrows: white—exopher, blue—muscle cell. Dashed lines mark embryos present in the uterus. Scale bar is 10 μm.

I    Offspring of worms with overactive exopheresis develop faster. Y.A.—young adult stage, L4—last larval stage. $n = 317–372$; $N = 2$.

Data information: Data are shown as mean ± SEM; $n$ represents number of worms; $N$ represents number of experimental repeats that were combined into a single value; *$P < 0.05$, **$P < 0.01$, ****$P < 0.0001$; (E, G) Mann–Whitney test; (F) two-tailed Welch's $t$-test; (I) Fisher's exact test.

Source data are available online for this figure.

*C. elegans* represents a previously uncharacterized nutrient management program associated with nourishing the next generation of progeny. We found that developing embryos in the uterus via embryo-derived factors trigger exopheresis, which reaches a maximum level between 2 and 4 days of adulthood. We observed that disturbance of yolk synthesis increases exopher biosynthesis, whereas inhibition of yolk uptake by oocytes reduces this process. Our results show that yolk protein produced in BWM is transferred to exophers and ultimately delivered to oocytes. Consequently, in mothers with highly active exopheresis, the volume of yolk content in embryos increased. On the contrary, food depletion intensifies exopher generation by the mother's BWM. Therefore, the use of exophers for the transport of vitellogenin represents an elegant mechanism by which remote cells can enrich the nourishment of developing embryos. This process leads to the production of larvae better prepared to thrive in the current environmental conditions. Thus, muscular exopheresis is likely an adaptive mechanism that affects the dynamics of population growth. The impact of exopheresis on early reproduction may be particularly crucial for wild worms, given the significantly shortened life expectancy observed under more natural conditions (Van Voorhies *et al*, 2005). In addition, yolk protein can be synthesized in the muscles of oviparous animals such as zebrafish (Zhong *et al*, 2014) and is supplemented from the mother to the intraovarian embryo in viviparous animals (Iida *et al*, 2019). Hence, it is tempting to speculate that the role of muscular exopheresis in supporting progeny development could be evolutionarily conserved. However, the exact mechanisms by which oocyte fertilization and subsequent embryonic development initiate exopher formation and how exopheresis is executed at the molecular level require further studies.

# Materials and Methods

## Reagents and Tools table

| Reagent or Resource | Source | Identifier |
|---|---|---|
| **Bacterial and Virus Strains** | | |
| *Escherichia coli* RNAi feeding strain | Caenorhabditis Genetics Center | HT115(DE3) |
| *E. coli* feeding strain | Caenorhabditis Genetics Center | OP50 |
| Ahringer RNAi library | Source BioScience | *C. elegans* RNAi Collection (Ahringer) |
| **Chemicals, Peptides, and Recombinant Proteins** | | |
| Bacto Agar | BioShop | AGR001.1 |
| Bacto Peptone | BioShop | PEP403.1 |
| Sodium Chloride | Chempur | 117941206 |
| Streptomycin sulfate salt | Sigma-Aldrich | S6501 |
| Nystatin suspension 10,000 unit/mL | Sigma-Aldrich | N1638 |
| Carbenicillin disodium salt | Roth | 6344.2 |
| Magnesium Sulfate heptahydrate | Sigma-Aldrich | M5921 |
| Potassium dihydrogen phosphate | Roth | 3904.1 |
| di-potassium hydrogen phosphate | Roth | P749.1 |
| Calcium Chloride (dihydrate) | BioShop | CCL444.500 |
| Cholesterol | BioShop | CHL500.5 |
| Sodium azide | Sigma-Aldrich | S2002 |
| Tetracycline HCl | BioShop | TET701 |
| IPTG | BioShop | IPT001 |
| Methylviologen dichloride hydrate | Sigma-Aldrich | 36541 |
| Polystyrene microspheres | Polysciences Europe GmbH | 08691-10 |
| Tetramisole hydrochloride | Sigma-Aldrich | L9756 |
| Paraformaldehyde | Sigma-Aldrich | 158127 |
| Hepes | Roth | HN77.3 |
| Glutaraldehyde | Sigma-Aldrich | 340855 |
| Ruthenium red | Sigma-Aldrich | R2751 |
| Gelatin | BioShop | GEL771 |
| Osmium tetroxide | Polysciences Europe GmbH | 0972B-5 |
| Potassium ferrocyanide | Sigma-Aldrich | P3289 |
| Thiocarbohydrazide | Sigma-Aldrich | 88535 |
| Hydrogen peroxide solution | Sigma-Aldrich | H1009 |
| 5-Fluoro-2'-deoxyuridine | Sigma-Aldrich | F0503 |
| Sodium hypochlorite solution 14% | VWR | 27900 |
| **Experimental Models: Organisms/Strains** | | |
| *C. elegans: wacIs6[myo-3p::pas-7::GGGGS Linker-wrmScarlet::unc-54 3′UTR, unc-119(+)], wacIs14 [myo-3p::tomm-20_1–50aa::attB5::mGFP::unc-54-3′UTR, unc-119(+)]* | This paper | Strain: ACH91 |
| *C. elegans: wacIs1[myo-3p::rpn-5 CAI = 0.97::GGGGS Linker-wrmScarlet::unc-54 3′UTR, unc-119(+)], wacIs14[myo-3 promoter::tomm-20_1–50aa::attB5::mGFP::unc-54-3′UTR, unc-119(+)]* | This paper | Strain: ACH93 |
| *C. elegans: wacIs1[myo-3p::rpn-5 CAI = 0.97::GGGGS Linker-wrmScarlet::unc-54 3′UTR, unc-119 (+)], vit-2(crg9070[vit-2::gfp]) X* | This paper | Strain: ACH199 |

**Reagents and Tools table**   (continued)

| Reagent or Resource | Source | Identifier |
|---|---|---|
| *C. elegans: rrf-3(b26) II; fem-1(hc17) IV; uthEx633 [myo-3p::GFP]* | Caenorhabditis Genetics Center | Strain: AGD885 |
| *C. elegans: wacIs1[myo-3 promoter::rpn-5 CAI = 0.97::GGGGS Linker-wrmScarlet::unc-54 3′UTR, unc-119(+)], wacIs14[myo-3 promoter::tomm-20_1–50aa::attB5::mGFP::unc-54-3′UTR, unc-119(+)], wwaEx2[unc-122 promoter::GFP]* | This paper | TUR5 |
| *C. elegans: him-5(e1490)V* | Caenorhabditis Genetics Center | Strain CB4088 |
| *C. elegans: zcIs14 [myo-3::GFP(mit)]* | Caenorhabditis Genetics Center | Strain SJ4103 |
| *C. elegans:* Bristol (N2) strain as wild-type (WT) | Caenorhabditis Genetics Center | Strain: N2 |
| **Recombinant DNA** | | |
| Plasmid: pCG150 (destination vector) | Addgene | #17247 |
| Plasmid: pMT26 (modified pCG150) | This paper | N/A |
| Plasmid: pSEM89 (wrmScarlet template) | Boulin Lab | El Mouridi *et al* (2017) |
| Plasmid: L1_Tomm20_1–55aa_R5 (tomm-20 1–55aa template) | This paper/ synthesized by Invitrogen GeneArt Gene Synthesis | N/A |
| Plasmid: L1_rpn-5_CAI097_R5 (rpn-5 CAI = 0.97 template) | This paper/ synthesized by Invitrogen GeneArt Gene Synthesis | N/A |
| Plasmid: L1_pas-7_R5 (pas-7 template)s | This paper/ synthesized by Invitrogen GeneArt Gene Synthesis | N/A |
| Plasmid: mGFP (mGFP template) | Bringmann Lab | N/A |
| Plasmid: pMT24 (pmyo-3::pas-7::wrmScarlet::unc-54-3′UTR in pCG150) | This paper | N/A |
| Plasmid: pMT28 (pmyo-3::tomm-20 1–55aa::mGFP::unc-54-3′UTR in pCG150) | This paper | N/A |
| Plasmid: pMT29 (pmyo-3::rpn-5 CAI = 0.97::wrmScarlet::unc-54-3′UTR in PCG150) | This paper | N/A |
| **Oligonucleotides** | | |
| For the list of oligonucleotides see Appendix Table S3 | N/A | N/A |
| **Software and Algorithms** | | |
| GraphPad Prism 8 | GraphPad Software, Inc. | N/A |
| ZEN | Zeiss | N/A |
| Office 365 | Microsoft | N/A |
| ImageJ | Wayne Rasband (NIH) | N/A |
| WormLab | MBF Bioscience | N/A |
| Leica Las X | Leica | N/A |
| Adobe Illustrator 2020 | Adobe | N/A |
| eC-CLEM | Open Source | N/A |

## Methods and Protocols

### Data reporting

No statistical methods were used to predetermine the sample size. The experiments were not randomized. The investigators were blinded to allocation during experiments and outcome assessment for exploratory behavior experiments and assays with embryos-conditioned buffer or lysed embryos and larvae.

### Worm maintenance and strains

Worms were maintained on nematode growth medium (NGM) plates seeded with OP50 *Escherichia coli* bacteria at 20°C unless otherwise stated (Brenner, 1974). A list of all strains used in the study, together with the information in which experiments they were used, is provided in Appendix Table S1.

### Generation of plasmids

All constructs were cloned using the SLiCE method (Zhang *et al*, 2012) and were sequenced for verification. The construct for mitochondrial outer membrane GFP expression in the BWM was prepared as follows. First, destination vector pMT26 containing pCG150 vector backbone with the *myo-3* promoter and *unc-54* 3′UTR sequences separated by a KspAI restriction site was constructed. Next, codon-optimized sequences (Redemann *et al*, 2011) for GFP, a linker containing the attB5 sequence, and the sequence of the first 55 amino acids from TOMM-20 protein (Watanabe *et al*, 2011) (from a plasmid that was synthesized for this study) were PCR amplified and inserted into linearized pMT26 vector (Appendix Table S3). To generate the construct for expression of RFP-tagged RPN-5 or PAS-7 in BWM, sequences encoding respective proteins were PCR amplified and inserted into pMT26-linearized

destination vector. As a template for the *rpn-5*-coding sequence, a plasmid bearing codon-optimized *rpn-5* cDNA with three artificial introns, which was synthesized for this study, was used. The sequence of *pas-7* was directly amplified from N2 genomic DNA, and the sequence of RFP (wrmScarlet) was amplified from the pSEM89 plasmid (El Mouridi *et al*, 2017).

### Transgenic strain generation

Transgenic strains were created by microparticle bombardment using *unc-119(ed3)* rescue as a selection marker (Praitis *et al*, 2001). After phenotypic confirmation of successful plasmid insertion, transformants were backcrossed two times against the N2 strain to remove the *unc-119(ed3)* background.

### Scoring exophers and fluorescence microscopy

For scoring of exophers, a confocal microscope or a stereomicroscope was used. When using the confocal microscope, animals were transferred onto 3% agarose pads prepared in $H_2O$ and formed on a microscope slide. Next, animals were immobilized on the pad using 6 µl of PolySciences 0.05 µm polystyrene microspheres or 25 µM tetramisole and covered with a glass coverslip. Immediately afterward, animals were imaged using an inverted Zeiss LSM800 laser-scanning confocal microscope with 40× or 63× oil immersion objectives. 488- and 561-nm lasers were used for excitation of the GFP and RFP fluorescent proteins, respectively. Z-stacks, which covered the whole animal, were collected, the number of exophers released by muscles was counted and compared between conditions.

For scoring exophers with the stereomicroscope, a Leica M165FC stereomicroscope equipped with Leica EL6000 lamp and standard Texas Red and GFP filter sets were used. Age-synchronized, freely moving day-2 adult animals were directly visualized on NGM plates, the number of visible exophers released by muscles was counted and compared between conditions.

The representative pictures of exophers presented in the manuscript were acquired using an inverted Zeiss 700 laser-scanning confocal microscope equipped with a 40× oil objective. 488- and 555-nm lasers were used to excite the GFP and RFP fluorescent proteins, respectively. To investigate the presence and distribution of exophers, Z-stacks were collected and processed with ZEN software.

### Electron microscopy

#### Sample preparation

*Caenorhabditis elegans* day-2 adult worms were fixed with 2% paraformaldehyde (Sigma-Aldrich, P6148) and 1% glutaraldehyde (Sigma-Aldrich, EM grade) in 0.2 M HEPES pH 7.3 overnight. Next, the samples were washed three times in 0.2 M Hepes pH 7.3, followed by overnight incubation with 0.1% ruthenium red solution in distilled water. After staining, worms were mounted in 2% gelatin on gridded 35mm dishes (MatTek 35mm dish, No. 1.5 Gridded Coverslip 14 mm Glass Diameter, P35G-1.5-14-CGRD) and designated for branding.

#### Near-infrared branding

Worms were localized on a gridded dish in bright field mode. Next, the red fluorescence in the exophers was imaged using a multiphoton Examiner.Z1 LSM 7MP microscope equipped with LSM NDD detectors and water 20× NA 1.0 objective (Zeiss, Oberkochen, Germany).

For the imaging, a pulsed laser at 1,020 nm (Coherent Chameleon) and 600–700 nm detection filter were used. Z-stacks of the samples were acquired using four times averaging in the line mode, pixel size of 138 × 138 nm and 500-nm interval in the Z-axis. For each sample, a Z-stack was acquired before and after the branding.

The ROI was marked by near-infrared branding (NIRB) (Bishop *et al*, 2011). The frame contour was bleached below the place of interest with a pulsed laser at 910 nm (Coherent Chameleon, 2 W at 910 nm) using 20–40 iterations with 20% of the laser power and pixel dwell time of 6–25 µs. The final parameters (number of iteration and pixel dwell time) were adjusted separately for each sample. The branding was repeated until the edges of the frame were highly fluorescent at 910 nm.

#### Processing

According to a published protocol, branded samples were prepared for electron microscopy (Deerinck *et al*, 2010) with minor changes (Śliwińska *et al*, 2020). Briefly, animals were post-fixed with a 1% aqueous solution of osmium tetroxide (Polysciences Europe GmbH 0972B-5) and 1.5% potassium ferrocyanide (Sigma-Aldrich, St. Louis, MO, USA, P3289) in phosphate buffer for 30 min on ice. Next, samples were immersed in 1% aqueous thiocarbohydrazide (Sigma-Aldrich, St. Louis, MO, USA, #88535) for 40 min, post-fixed with a 2% aqueous solution of osmium tetroxide for 60 min (all at room temperature), and incubated in 1% aqueous uranyl acetate at 4°C overnight. The next day, samples were immersed in 0.66% lead aspartate for 60 min at 60°C, dehydrated with increasing ethanol dilutions, infiltrated with Durcupan resin (Sigma-Aldrich, St. Louis, MO, USA, #44610), embedded using BEEM capsules according to a published protocol (Hanson *et al*, 2010), and cured at 70°C for 72 h.

The resin blocks were trimmed and cut with an ultramicrotome (ultracut R or EM UC7, Leica), and ultrathin sections (70 nm thick) were collected on formvar-coated copper grids, mesh 100 (Agar Scientific, AGS138-1 or EMS FF100-CU-50), or copper slot grids (EMS FCF 2010-CU-SB-50). For serial block-face scanning electron microscopy, trimmed pyramids were cut off with a razor blade, mounted to aluminum pins (Gatan system pins, Micro to Nano, Netherlands, 10-006003-50) with cyanoacrylate glue, trimmed, block-face polished, and grounded with silver paint (Ted Pella, Redding, USA, 16 062-15).

#### Transmission electron microscopy

Specimen grids were examined with a JEM 1400 transmission electron microscope (JEOL Co., Tokyo, Japan, 2008), equipped with a MORADA G2 11-megapixel TEM camera (EMSIS GmbH, Münster, Germany) at Nencki Institute of Experimental Biology PAS or with a Tecnai T12 BioTwin electron microscope (FEI, Hillsboro, OR, USA) equipped with a 16 megapixel TemCam-F416 (R) camera (TVIPS GmbH) at the International Institute of Molecular and Cell Biology in Warsaw.

#### Exopher tracking

Identification of branded exophers was performed with eC-CLEM open-source software (Paul-Gilloteaux *et al*, 2017). The NIRB frame is visible on both FM and EM images. The marks on the EM image are placed on the frame's edges and, respectively, its position is adjusted on the FM image allowing for correlation of both images.

These coordinates allow the setting of exopher position on the TEM image.

### RNA interference

RNA interference in *C. elegans* was performed using the standard RNAi feeding method and RNAi clone (Kamath & Ahringer, 2003). For experiments, NGM plates supplemented with 1 mM IPTG and 25 μg/μl carbenicillin seeded with HT115 *E. coli* bacteria expressing double-stranded RNA (dsRNA) against the gene of interest or, as a control, bacteria with the empty vector were used. Worms were placed on freshly prepared RNAi plates, either as age-synchronized pretzel-stage embryos, L1 larvae, or L4 larvae. The number of exophers was counted in day-2 adult worms using a confocal microscope or a stereomicroscope.

### Stress influence on exopher production

As previously described, worms were age-synchronized using alkaline hypochlorite solution (bleaching procedure) (Porta-de-la-Riva *et al*, 2012). The harvested embryos were incubated overnight at 16°C for hatching. Approximately 1,000 L1 larvae were transferred to NGM plates and incubated at 20°C till they reached day 2 of adulthood. The worms were further channeled to the respective stress treatments.

### Oxidative stress

Approximately 100 day-2 adult worms were washed from the NGM plates and rinsed 3 times with M9 buffer. The worms to be stressed were suspended in 1 ml of 5 mM hydrogen peroxide solution prepared in M9 buffer, whereas control worms were suspended in M9 buffer. The tubes were incubated on a shaker at 20°C for 60 min.

### Heat stress

Similarly, approximately 100 day-2 adult worms were washed from NGM plates and rinsed three times with M9 buffer. The worms were further suspended in a 1 ml M9 buffer. The worms to be heat stressed were incubated in a shaker at 33°C for 60 min, whereas the control animals were incubated at 20°C for 60 min.

### Exopher quantification

From each stress/control treatment, 30 worms were picked onto agarose pad slides individually for exopher quantification. Exophers were quantified using a confocal microscope. Obtained data were compared between conditions.

### Number of exophers on consecutive days

For each biological replicate, 30 L4 larvae hermaphrodites or 50 L4 larvae males were transferred to fresh NGM plates (five hermaphrodites or 10 males per plate). For the next consecutive 15 days, the number of exophers in each animal was counted using a stereomicroscope. Hermaphrodites were transferred to fresh plates every 2–3 days. Males were kept on the same plate until the end of the experiment. All animals that died during the experiment time course were removed from the plate.

### Measuring exophers in fem-1 mutant

Approximately 50 L1 larvae from *fem-1(hc17)ts* mutant strain expressing GFP in BWM were transferred to two fresh NGM plates. During the time course of the experiment, one of the plates was kept at 15°C while the second one was kept at 25°C. When worms reached L4 larval stage, each worm was transferred to a separate plate and grown at the same temperature as before. After 48 h at 25°C or 72 h at 15°C, using a stereomicroscope, a number of exophers released from worms' muscles were counted and plates were scored for F1 offspring to assign all worms as fertile or infertile.

Additionally, a phenotype rescue experiment was performed for sterile animals grown at 25°C. 24 h after they reached adulthood, single animals were placed on separate plates and mated with 6–8 *him-5* mutant day-1 males. After the next 48 h, a number of exophers released from the muscles were counted.

### FUdR assay

Age-synchronized animals were placed on NGM plates seeded with OP50 *E. coli* bacteria as a food source until they reached young adulthood (day 0). Next, animals were selected and moved to test plates containing 25 μM fluorodeoxyuridine (FUdR) to prevent embryonic development and egg hatching (Mitchell *et al*, 1979) or control plates without FUdR. The number of exophers was counted on adult day 2 using confocal microscopy.

### Exopher and in utero embryo correlation

For correlating the number of exophers with the number of embryos present in worms' uterus, 1- to 3-day-old animals were used. First, muscular exophers for a single worm were counted using the stereomicroscope. Next, a worm was transferred to a 10-μl drop of 1.8% hypochlorite solution on a microscope slide. Finally, after approximately 5 min, when the hermaphrodite mother was entirely disrupted, the number of embryos released from the uterus was counted. Data analysis was performed using GraphPad Prism 8 software.

### Starvation assay

To assess the influence of worms' starvation on exopher production, day-2 adult worms and L4 larvae were moved to bacteria-free NGM plates. After 24 h of food deprivation, the number of exophers was counted using a stereomicroscope. As a control, day-3 adult worms and day-1 adult worms grown for the entire time on bacteria-seeded NGM plates were used. Obtained data were compared between conditions.

### Embryo lysate assay

To obtain embryos, age-synchronized N2 gravid hermaphrodites were disrupted in 1.8% hypochlorite solution. Harvested embryos were suspended in an appropriate volume of M9 buffer to reach the concentration of approximately 200 embryos per μl. As a control, age-synchronized L2/L3 larvae were collected and suspended in M9 buffer to reach the concentration of approximately 25 larvae per μl. Next, embryos and larvae were flash-frozen in liquid nitrogen, thawed on ice, and sonicated three times for 10 s to obtain embryos and larvae lysate. 150 μl of embryos or larvae lysate was mixed with 150 μl of concentrated OP50 *E. coli* bacteria and placed in a 0.5-ml Eppendorf tube. Approximately 30 1-day-old young adult hermaphrodites or males were transferred to Eppendorf tubes containing a mixture of embryos or larvae lysate and bacteria. Next, the Eppendorf tube was placed on the rotator for 1 h at room temperature. After 1 h, the tube contents were placed on a fresh NGM plate seeded with OP50 bacteria. 18 h later, the number of

exophers in each worm was counted. The entire protocol was the same for mock control animals except that no worms were used in the hypochlorite treatment step.

### Assay with embryo-conditioned buffer

To obtain embryos, age-synchronized N2 gravid hermaphrodites were disrupted in 1.8% hypochlorite solution. Harvested embryos were suspended in an appropriate volume of M9 buffer to reach a concentration of approximately 200 embryos per µl. 150 µl of embryos were mixed with 150 µl of concentrated OP50 *E. coli* bacteria and placed in a 0.5-ml Eppendorf tube. 30–50 day-1 young adult hermaphrodites were transferred to Eppendorf tubes containing a mixture of embryos and bacteria. Next, the Eppendorf tubes were placed on the rotator for 1 h at room temperature. After 1 h, the contents of the tubes were placed on fresh NGM plates seeded with OP50 bacteria. Approximately 20 h later, the number of exophers in each worm was counted. The entire protocol was the same for mock control animals except that no worms were used in the hypochlorite treatment step.

### Worm exploratory behavior

Age-synchronized day-2 adult worms which had an average (6–8), few (< 2), or many (> 20) exophers were sorted to separate NGM plates. Approximately 10 worms per replicate were placed onto NGM plates, and worm movement was recorded for 2 min using the WormLab system (MBF Bioscience). The frame rate, exposure time, and gain were set to 7.5 frames per second, 0.0031 s, and 1, respectively. The track length, straight-line distance, center point speed, and the overall track pattern of individual worms were analyzed using the WormLab software (MBF Bioscience).

### Vitellogenin levels in embryos

Vitellogenin levels in embryos were measured based on the GFP signal from fluorescently tagged VIT-2 protein. Approximately 200 late L4 larvae were transferred to a fresh NGM plate. On day 2 of adulthood, the number of exophers in each animal was counted, and worms with less than two or more than 20 visible exophers were transferred to new plates. Next, animals from each group were individually transferred to a 10 µl drop of M9 buffer placed on a microscope slide. Using a sharp injection needle, each worm was cut open to release the embryos from the uterus. Fluorescent signal from the 2-cell embryo stage was captured using a Leica M165FC stereomicroscope equipped with Leica EL6000 lamp, a standard GFP filter set, and Leica DFC365 FX CCD camera. The magnification used for image collection was set to 192x. Exposure time and gain were set to 600 ms and 2, respectively. The fluorescent signal was quantified using Leica Las X software and was normalized to the average signal from embryos obtained from animals with less than two exophers.

To measure vitellogenin levels in embryos after *ced-1* knockdown, the entire procedure was the same except that day-2 adult worms were picked randomly and as a control, animals were used that were grown on plates seeded with bacteria containing the empty vector.

### Worm development assay

25–30 age-synchronized day-2 adult worms with an average (6–8), few (< 2), or many (> 20) exophers were sorted to separate NGM plates. Gravid adults were allowed to lay eggs for 4 h and then removed from the plates, and the development of their offspring was followed. 46 h later, using the stereomicroscope, the developmental stage of each animal was checked, and the proportion between L4 larvae stage worms and young adult worms was calculated.

### Statistical analysis

Statistical tests used in this study were the Mann–Whitney test and Fisher's exact test. *P*-value < 0.05 was considered significant.

# Data availability

This study includes no data deposited in external repositories.

**Expanded View** for this article is available online.

## Acknowledgements

We thank the *Caenorhabditis* Genetics Center (funded by the NIH National Center for Research Resources, P40 OD010440) for strains; T. Hoppe for expert advice and Carl Kutzner for RNAi clones; T. Wegierski for confocal microscopy assistance; B. Uszczyńska-Ratajczak, K. Szczepanowska, H. Bringmann, and members of Chacińska and Pokrzywa laboratories for discussions and comments on the manuscript. Multiphoton microscopy imaging, near-infrared branding, and serial block-face scanning electron microscopy imaging were performed at the Laboratory of Imaging Tissue Structure and Function at Nencki Institute of Experimental Biology PAS. Transmission electron microscopy imaging and ultramicrotomy were performed either at the Core Facility of the International Institute of Molecular and Cell Biology in Warsaw or using the equipment of the Laboratory of Electron Microscopy at the Nencki Institute of Experimental Biology PAS. Graphical abstract created with BioRender.com. Work in the W.P. laboratory was mainly funded by the Foundation for Polish Science co-financed by the European Union under the European Regional Development Fund (grant POIR.04.04.00-00-5EAB/18-00 to K.B., K.K. and W.P.) and additionally supported by the European Molecular Biology Organization (EMBO Installation Grant No. 3916 to W.P.) and the Deutsche Forschungsgemeinschaft (DFG FOR 2743 to W.P.). The research leading to part of these results has received funding from the Norwegian Financial Mechanism 2014-2021 and operated by the Polish National Science Center under the project contract no UMO-2019/34/H/NZ3/00691 (M.P.). Work in the A.C. laboratory was funded by the "Regenerative Mechanisms for Health" project MAB/2017/2 (A.C. and M.T.) carried out within the International Research Agendas programme of the Foundation for Polish Science co-financed by the European Union under the European Regional Development Fund and was supported by a POLONEZ Fellowship of National Science Centre, Poland, 2016/21/P/NZ3/03891 (M.T.), within European Union's Horizon 2020 research and innovation program under the Marie Skłodowska-Curie grant agreement no. 665778. Work in the M. T. Laboratory was funded by a National Science Centre SONATA grant (2019/35/D/NZ3/04091).

## Author contributions

Conceptualization: MT, WP; Data curation: MT, WP; Formal analysis: MT, KB, MP, NS; Funding acquisition: WP, AC, MT; Investigation: MT, WP, KB, MP, NS, MM, MAŚ, MN, KK, NN; Methodology: MT, WP, MM, MAŚ; Project administration: WP, AC, MT; Resources: WP, AC, MT; Supervision: WP, AC; Validation: MT, WP; Visualization: MT, WP, KB, NS, MM, MAŚ; Writing – original draft: WP, MT; Writing – review & editing: WP, MT, AC.

## Conflict of interest

The authors declare that they have no conflict of interest.

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
