## [Review Process File · EMBO Reports]

Muscle-derived exophers promote reproductive fitness

Michał Turek, Katarzyna Banasiak, Małgorzata Piechota, Nilesh Shanmugam, Matylda Macias, Małgorzata Śliwińska, Marta Niklewicz, Konrad Kowalski, Natalia Nowak, Agnieszka Chacinska, and Wojciech Pokrzywa

DOI: [10.15252/embr.202052071](https://doi.org/10.15252/embr.202052071)

Corresponding author(s): [Wojciech Pokrzywa \(wpokrzywa@iimcb.gov.pl\)](mailto:Wojciech.Pokrzywa@iimcb.gov.pl), [Michał Turek \(m.turek@ibb.waw.pl\)](mailto:Michał.Turek@ibb.waw.pl)

Review Timeline:

Submission Date:	11th Nov 20
Editorial Decision:	1st Dec 20
Revision Received:	30th Mar 21
Editorial Decision:	29th Apr 21
Revision Received:	8th May 21
Accepted:	21st May 21

Editor: *Martina Rembold*

Transaction Report:

Dear Wojtek,

Thank you for the submission of your research manuscript to our journal. We have now received the full set of referee reports that is copied below.

As you will see, the referees acknowledge that the findings are potentially interesting. However, they also raise a number of largely overlapping concerns and have a number of constructive suggestions to further strengthen and substantiate your findings. Regarding the postulated secreted factor that induces exophoresis, all suggested experiments should be performed (repeat experiments in a double-blinded fashion, test embryo-conditioned medium, test it on male worms etc) but it will not be required to determine the factor that is secreted. Alternative explanations, such as mechanical cues or eggshell components should also be taken into account, at least in the interpretation of the data and in the discussion.

Given these constructive comments, we would like to invite you to revise your manuscript with the understanding that the referee concerns (as detailed above and in their reports) must be fully addressed and their suggestions taken on board. Please address all referee concerns in a complete point-by-point response. Acceptance of the manuscript will depend on a positive outcome of a second round of review. It is EMBO reports policy to allow a single round of revision only and acceptance or rejection of the manuscript will therefore depend on the completeness of your responses included in the next, final version of the manuscript.

We invite you to submit your manuscript within three months of a request for revision. This would be March 1st, 2021 in your case. However, we are aware of the fact that many laboratories are not fully functional due to COVID-19 related shutdowns and we have therefore extended the revision time for all research manuscripts under our scooping protection to allow for the extra time required to address essential experimental issues. Please contact us if you wish to discuss the time needed and the revisions further.

- 1) A data availability section is missing.
- 2) Your manuscript contains error bars based on $n=2$. Please use scatter blots showing the individual datapoints in these cases. The use of statistical tests needs to be justified.

When submitting your revised manuscript, please carefully review the instructions that follow below. Failure to include requested items will delay the evaluation of your revision.*****

- 1) a .docx formatted version of the manuscript text (including legends for main figures, EV figures and tables). Please make sure that the changes are highlighted to be clearly visible.
- 2) individual production quality figure files as .eps, .tif, .jpg (one file per figure).

Please download our Figure Preparation Guidelines (figure preparation pdf) from our Author Guidelines pages

<https://www.embopress.org/page/journal/14693178/authorguide> for more info on how to prepare your figures.

4) a complete author checklist, which you can download from our author guidelines (). Please insert information in the checklist that is also reflected in the manuscript. The completed author checklist will also be part of the RPF.

5) Please note that all corresponding authors are required to supply an ORCID ID for their name upon submission of a revised manuscript (). Please find instructions on how to link your ORCID ID to your account in our manuscript tracking system in our Author guidelines
()

6) We replaced Supplementary Information with Expanded View (EV) Figures and Tables that are collapsible/expandable online. A maximum of 5 EV Figures can be typeset. EV Figures should be cited as 'Figure EV1, Figure EV2" etc... in the text and their respective legends should be included in the main text after the legends of regular figures.

7) Please note that a Data Availability section at the end of Materials and Methods is now mandatory. In case you have no data that requires deposition in a public database, please state so instead of refereeing to the database.

See also < <https://www.embopress.org/page/journal/14693178/authorguide#dataavailability>>).

Please note that the Data Availability Section is restricted to new primary data that are part of this study.

8) We would also encourage you to include the source data for figure panels that show essential data. Numerical data should be provided as individual .xls or .csv files (including a tab describing the data). For blots or microscopy, uncropped images should be submitted (using a zip archive if multiple images need to be supplied for one panel). Additional information on source data and instruction on how to label the files are available .

9) Our journal encourages inclusion of *data citations in the reference list* to directly cite datasets that were re-used and obtained from public databases. Data citations in the article text are distinct from normal bibliographical citations and should directly link to the database records from which the

data can be accessed. In the main text, data citations are formatted as follows: "Data ref: Smith et al, 2001" or "Data ref: NCBI Sequence Read Archive PRJNA342805, 2017". In the Reference list, data citations must be labeled with "[DATASET]". A data reference must provide the database name, accession number/identifiers and a resolvable link to the landing page from which the data can be accessed at the end of the reference. Further instructions are available at .

10) Regarding data quantification

- the name of the statistical test used to generate error bars and P values,
- the number (n) of independent experiments (please specify technical or biological replicates) underlying each data point,
- the nature of the bars and error bars (s.d., s.e.m.)

11) As part of the EMBO publication's Transparent Editorial Process, EMBO reports publishes online a Review Process File to accompany accepted manuscripts. This File will be published in conjunction with your paper and will include the referee reports, your point-by-point response and all pertinent correspondence relating to the manuscript.

I look forward to seeing a revised version of your manuscript when it is ready. Please let me know if you have questions or comments regarding the revision.

Kind regards,

Martina

Referee #1:

Very little is known about the role of exophers, large membrane-enclosed vesicles that are extruded from cells. Exophers were first identified in the worm *C. elegans*, which uses exophers to remove damaged, degraded and/or aggregated materials and dysfunctional mitochondria from neurons. Recently, a similar process has been observed in cardiomyocytes. There are some really interesting results in this paper. Here, the authors describe exopher production by *C. elegans* muscle cells, and

convincingly show that embryo presence is correlated with exopher production. The muscle-derived exophers appear to be a source of yolk protein for developing oocytes. The authors claim that something secreted by the embryos induces muscle cell exopher production. This is surprising and not strongly supported. I recommend repeating a few key experiments in a double-blind fashion to reduce potential bias and I also make a few suggestions for additional experiments and/or edits that would strengthen the paper.

Minor:

Include a brief introduction to how yolk usually gets into oocytes.

74. 'probably not compromised' might be better as 'apparently intact'.

79. If EMB-8 and POD-1 play a specific role in exopheresis, it would be better not to refer to them as polarity regulators. Otherwise, how exopheresis relates to polarity regulation should be described.

117 I recommend keeping the interpretations closer to the data. Fig. 2E does not characterize these external vesicles as exophers or show that males are 'devoid of mechanisms' that trigger expulsion.

172. "maturing eggs" should be 'maturing oocytes'. The yolk gets into the oocytes, not the eggs. It would be better if the authors avoided the word 'egg', instead being careful throughout to use the terms 'oocytes' for unfertilized and 'embryos' for fertilized.

192. 'yolk-reach' should be 'yolk-rich'.

196. Please reword/clarify the confusing sentence beginning "Accordingly, based on their content and participation of the autophagy machinery in exophers formation..."

203-204. Reword sentence for clarity and to reduce wordiness.

Fig. 2A Clarify number of exophers per what?

Fig. 2B. Clarify why fem-1 'offspring +' have fewer exophers than fem-1 'offspring -'. Is this in error?

Fig. 4 Showing oocytes that have taken up exopher-provided GFP yolk proteins would be a nice addition.

114 (etc.)

I am not so sure that developing embryos per se induce muscular exopheresis (i.e. do they need to be 'developing'? Or is presence enough?). The data in Fig. 3 A-C is consistent with the cells of the uterus inducing the exopheresis, perhaps by detecting the presence of the eggs through stretch gated ion channels. But, then the authors show embryonic EXTRACT can induce exopheresis! This is very surprising, and based on a result (Fig. 3E) that is not, statistically, very strong. The experiments were not done blind, which adds to the concern of possible unintentional bias. I would like to see this experiment repeated in a double-blind fashion and a potential mechanism for how something in egg lysate communicates to the muscle cells proposed. Are extracts able to induce exopheresis in males? If so, you might have a nice assay to identify this mystery factor. If not, I would revisit the possibility that egg sensing by the uterine cells underlies the effect seen.

139 The authors state that 'molecules that diffuse from embryos' induce exophoresis. In addition to the comments above, the experiment (adding sonicated embryo lysate) does not rule out a role for eggshell components.

140. Here and elsewhere, it would be better to reframe the introductory statements in neutral way, rather than looking for a preferred conclusion. For example, the sentence "Finally, we decided to confirm that even in the face of disturbed proteostasis specifically in the BWM, a signal associated with the developing embryos in the uterus would be the primary regulator of muscular exophoresis" would be better rephrased as: "Finally, we explored the role of BWM proteostasis in exopher formation".

140-150. I question the interpretation of the experiment shown in Fig. 3F. The authors show depletion of UNC-45 by RNAi decreases exopher numbers. This could easily be due to disruption of an acto-myosin based exopher extrusion mechanism in the muscle cells. Therefore, this data does not show the 'predominant role of maturing eggs in muscular exophoresis'. (minor: delete 'maturing', replace 'eggs' with 'embryos').

197. The authors claim that the autophagy machinery is involved in exopher production but do not perform experiments directly addressing this claim. Additional experiments that explore the role of autophagy in exopher production would strengthen the paper.

Referee #2:

In this study entitled "Nutritional status and fecundity are synchronised by muscular exophoresis", Turek et al. show that muscle cells expel exophers, large extracellular vesicles, to provide energy resources for the next generation. Yolk protein is secreted within exophers, which nourishes the embryos in utero and even later outside the mother. Such a pinch-off of a large part of the cytosol has been first observed by Melentijevic et al. in *C. elegans* neurons, which has been attributed to a proteostasis pathway to remove damaged and dysfunctional organelles and macromolecules from the cell. In contrast, exophoresis in muscle cells seems to have less of a stress relief purpose than a growth and reproductive purpose. This is a fascinating discovery that is of interest to a broad readership. Having said this, there are still a few points that need to be addressed:

Major points:

1. The authors need to give the exact absolute numbers of worms with exophers and exophers per worm rather than the "fold change" of exophers. Absolute numbers are much more informative, and since this is the first time this phenomenon has been characterized, it is important that the reader can estimate the frequency of these events.
2. Some exophers are very similar to coelomocytes (e.g., Fig. 1F right panel with white arrow) and therefore it would be very helpful if the authors could show how they compare to each other (size, location, content?) and how the authors distinguish exophers from coelomocytes in their analysis.
3. The numbers of exophers or worms with exophers seem to be surprisingly variable between experiments. For example: Fig. 2A: here, the data indicate that on day 2 and 3 of adulthood, there are approx. 6-9 exophers per worm, with little variation. In contrast, the graph in Fig. 2D indicates that 30% of worms have more than 10 exophers. If this were the case, then the error bars in Fig. 2A would be much larger. Is there an explanation for this discrepancy?

4. Fig. 2: The authors claim that exopher secretion is sex specific. However, males do not produce eggs and since these trigger exopher release, the trigger could simply be missing. Therefore, the authors should test whether embryo lysate is able to trigger exopheresis in males.
5. Fig. 3A: according to the literature (and our own observations) hermaphrodites have usually 10-15 eggs in the uterus. The graph shows numbers of 25 up to 50, and I cannot even imagine what a worm with this high number of eggs would look like... Also the description "animals were 1-3 days old" is much too imprecise because the age in this range drastically influences the number of exophers (see Fig. 2A).
6. Fig. 3F: There are too many variables in this experiment, which does not allow any conclusion at the moment, since *unc-45* RNAi not only reduces proteostasis but also affects the development of the embryos. The authors need to further dissect this and treat animals with *unc-45* RNAi in the presence of egg lysates. This way, they could separate the effect on muscle proteostasis from a potential negative growth signal of the affected embryos.
7. Fig. 4H: It is surprising that the VIT-2::GFP is highly abundant in embryos, but no RPN-5::RFP is detected in the embryos. If the entire content of the exophers were taken up by embryos one would expect to detect both, RFP and GFP staining. Is there a selective uptake of exopher content? How could that happen? Alternatively, one could imagine that the transfer of VIT-2 is mediated by another parallel pathway, distinct from exophers...

Minor points:

1. Fig. 1 legend (I): The description "proteostasis induced via oxidative stress [...]" is not correct. Please rephrase into "Challenging proteostasis via [...]".
2. Fig. 2B: labeling is wrong (+/- offspring).
3. Fig. S2: The title misleading and should rather read "Active exopheresis does NOT affect worm motility".
4. The authors misspell "exopheresis" with "exophoresis" several times throughout the text (e.g. in the title of Fig. S2, see above). Please correct.
5. Line 162: "exopher" should be plural.

Referee #3:

Turek, Piechota et al. demonstrate that body wall muscle releases 2-15 μm cell fragments called exophers containing proteasomes, yolk proteins, and damaged mitochondria. Similar structures were previously reported to export waste for neurons and cardiac muscle, but the authors provide evidence that there are different subpopulations of exophers and that many body wall muscle exophers have trophic functions. They find that exopher production is regulated by female fertility or embryo production, while embryos seem to benefit nutritionally from increased exopher production. Given the unpublished report of exophers released from skeletal muscle in mice, these findings in *C. elegans* are likely to be conserved in mammals and provide important new insight into organismal physiology.

Major points

Is the right panel of Fig. 1F showing a coelomocyte? It looks like the marker has been endocytosed by a round cell and is found in many endosomes. It does not look consistent with exopher fragmentation as the authors describe in the text. Also, if the two panels are different time points, the intervening time should be labeled on the figure or given in the figure legend.

Fig 1H-I, 2B, 2D, 3B-C, 3E-F, S1B, S3A - Data should be presented as absolute numbers of exophers per animal like Fig. S1A, not fold change. This is especially relevant to see how exopher number in

the various classes of fem-1 mutants compare to control animals.

Also, it is important to consider that a change in exopher number can be a result of a change in exopher production, exopher clearance, or both. Therefore it is important to report data throughout the manuscript in terms of exopher number, not exopher production.

Fig. 3F is missing a control that exopher production by muscle cells is not altered by paralysis. For example, *unc-54* RNAi could be used as a control targeting myosin that does not disrupt embryonic development.

How do the authors propose that vitellogenins in 2-15 μm exophers are being fed to oocytes if not through exopher phagocytosis. Would a *ced-1* mutant strain still show the increase in VIT-2::GFP in high exopher worms?

Minor points

Fig. S1A - Would be better represented as paired data showing the number of proteasome-containing exophers and the corresponding number of mitochondria-containing exophers in each animal.

73 - "probably not compromised mitochondria"

Does this mean the mitochondria had normal appearing cristae? In other exopher studies, they tested mitochondrial function by looking at oxidation or other functional assays. Either reword to describe the assay or test mitochondrial function.

Fig. 2B - Offspring label is inconsistent with the text.

Line 115-118 - The text is not clear how the authors were scoring exophers vs "exopher-like" vs "non-extruded exophers". What is the distinction in Fig. 2E? Are these images projections or a single focus in the plane of the muscle?

Fig. 3A - 2-3 day old adults hold more embryos in the uterus than young adults and the authors showed in Fig. 2A that exophers peak at day 2-3 of adulthood. This analysis should not include data from day 1 adults to avoid conflating the effect of holding eggs with developmental timing.

Could the authors briefly discuss what is known about the effect of egg-laying mutants and starvation on oocyte production, maturation, or ovulation? The observed effects on exophers could be tertiary to secondary effects on germ cell biology. Do strains with increased germ cell divisions that make germ line tumors, such as *glp-1(gof)*, have altered exopher numbers?

Lysed embryos in Fig. 3D-E are not an approximation for molecules diffusing from developing embryos. Are the authors proposing a small molecule signal that is able to cross the permeability barrier of the eggshell? Conditioned media incubated with embryos would be more comparable. Either repeat experiment or rephrase. It may also be worth considering the mechanical impact on the body wall muscle of holding embryos in the uterus in *egl* mutants, etc.

Line 197- The authors show no data for autophagy machinery. Missing reference(s)?

Line 545 - How do the HT115 bacteria without a vector grow on carbenicillin plates? Describe RNAi control condition more clearly.

English editing is required. For example, "uptaken" is not a verb. The correct phrase is "taken up".

Dr. Martina Rembold
Senior Editor
EMBO reports

Wojciech Pokrzywa, PhD
Head of the Laboratory of Protein Metabolism
International Institute of Molecular and Cell
Biology
4 Ks. Trojdena Street
02-109 Warsaw
Poland
wpokrzywa@iimcb.gov.pl
+48225970743

Dear Dr. Martina Rembold

Thank you for the positive response on our manuscript "*Nutritional status and fecundity are synchronised by muscular exopheresis*" (MBOR-2020-52071V1). We are pleased to read that the reviewers found our results interesting, fascinating, and important and that they recognized that the manuscript is well written and will be interesting for a wide audience. In response to reviewers' comments, significant experimental work was undertaken, the results of which are shown in new panels in figures 1D, 1H, 2D, 3E-F, EV1A, EV2A-D, EV3B, EV4B and Appendix Table 1. In addition, we have made all suggested changes to the text, analyses, and calculations. We thank the reviewers for constructive comments that improved our manuscript, and we are happy to present this revised version.

Please find a detailed description of the edited paragraphs below (the reviewers' comments are **bold** and our responses are *italic*):

Reviewer: 1

Comments to the Authors

Very little is known about the role of exophers, large membrane-enclosed vesicles that are extruded from cells. Exophers were first identified in the worm *C. elegans*, which uses exophers to remove damaged, degraded and/or aggregated materials and dysfunctional mitochondria from neurons. Recently, a similar process has been observed in cardiomyocytes. There are some really interesting results in this paper. Here, the authors describe exopher production by *C. elegans* muscle cells, and convincingly show that embryo presence is correlated with exopher production. The muscle-derived exophers appear to be a source of yolk protein for developing oocytes. The authors claim that something secreted by the embryos induces muscle cell exopher production. This is surprising and not strongly supported. I recommend repeating a few key experiments in a double-blind fashion to reduce potential bias and I also make a few suggestions for additional experiments and/or edits that would strengthen the paper.

Minor:

1. Include a brief introduction to how yolk usually gets into oocytes.

This brief introduction "Yolk is transported to proximal oocytes via RME-2 receptor-mediated endocytosis (Grant and Hirsh, 1999; Hall et al., 1999)" is now included in the section on the effect of RME-2 RNAi on exophers generation (page...)

2. 'probably not compromised' might be better as 'apparently intact'.

We have made appropriate changes in the text.

3. If EMB-8 and POD-1 play a specific role in exoheresis, it would be better not to refer to them as polarity regulators. Otherwise, how exoheresis relates to polarity regulation should be described.

We have made appropriate changes in the text.

4. I recommend keeping the interpretations closer to the data. Fig. 2E does not characterize these external vesicles as exophers or show that males are 'devoid of mechanisms' that trigger expulsion.

In the main text and legend to Figure 2E, we noted that exopher marker-labeled structures that resemble exophers (size/shape) were not excreted by the BWM of FUDR-treated hermaphrodites and males. This observation potentially indicates an active process of segregation and storage of material that constitutes the natural contents of exophers, but not an active mechanism for their excretion outside the muscle. We have, therefore, suggested that males are 'devoid of mechanisms' that trigger exopher expulsion. However, we understand that we might have too hastily characterized these intramuscular objects as non-secreted exophers. We have now rephrased the statement to "We also found that hermaphrodites treated with FUDR often contained structures in their BWM that appeared like segregated exopher cargo (Figure 2E, middle and right panels). Interestingly, we detected similar objects in males (Figure 2E, left panel) that, as in sterile hermaphrodites, are not excreted by the BWM."

5. "maturing eggs" should be 'maturing oocytes'. The yolk gets into the oocytes, not the eggs. It would be better if the authors avoided the word 'egg', instead being careful throughout to use the terms 'oocytes' for unfertilized and 'embryos' for fertilized.

We have made appropriate changes in the text.

6. 'yolk-reach' should be 'yolk-rich'.

We have made appropriate changes in the text.

7. Please reword/clarify the confusing sentence beginning "Accordingly, based on their content and participation of the autophagy machinery in exophers formation..."

The text has been changed accordingly: “Recent reports indicate that the class of the largest extracellular vesicles, known as exophers, are responsible for removing neurotoxic components in neurons (Melentijevic et al., 2017) and damaged mitochondria in cardiomyocytes (Nicolas-Avila et al., 2020). Here we show that exophers are not only a storage compartment for cellular waste, but that the muscular exopheresis in C. elegans...”

8. 203-204. Reword sentence for clarity and to reduce wordiness.

The text has been edited for clarity. The current sentence is: “Recent reports indicate that the class of the largest extracellular vesicles, known as exophers, are responsible for removing neurotoxic components in neurons (Melentijevic et al., 2017) and damaged mitochondria in cardiomyocytes (Nicolas-Avila et al., 2020). Here we show that exophers are not only a storage compartment for cellular waste, but that the muscular exopheresis in C. elegans represents a previously uncharacterized nutrient management program associated with nourishing the next generation of progeny. We found that developing embryos in the uterus trigger exopheresis, which reaches a maximum level between 2 to 4 days of adulthood.”

9. Fig. 2A Clarify number of exophers per what?

We have clarified the Y axis labelling for all panels showing number of exophers. Now they are labelled as “No. of exophers/animal”.

10. Fig. 2B. Clarify why fem-1 'offspring +' have fewer exophers than fem-1 'offspring -'. Is this in error?

Yes, it was a typing error, and the figure was marked correctly.

11. Fig. 4 Showing oocytes that have taken up exopher-provided GFP yolk proteins would be a nice addition.

We agree with the reviewer that showing delivery of the muscle vitellogenin to the oocytes would be a nice addition. We were very keen on this result and therefore spent the last months generating a worm line in which fluorescently labelled VIT-2 would be expressed from the myo-3 promoter. This would allow us to follow the trafficking of muscle-derived vitellogenin, hoping to visualize it in oocytes. We generated large (11 kb) Pmyo-3::vit-2::GFP DNA constructs that we intended to introduce into worms in two ways. Single copy insertion via CRISPR/Cas-9-based SKI LODGE system (Silva-Garcia et al., 2019) and microinjection with the created construct. Unfortunately, our several rounds of microinjections with vit-2::GFP template DNA for SKI LODGE system, injected together with rol-6 marker, did not lead to any animals with the roller phenotype. Moreover, microinjections of the Pmyo-3::vit-2::GFP DNA construct also did not result in any animal expressing vit-2::GFP. We injected it into two different worm strains, unc-119(ed3) mutants (Pmyo-3::vit-2::GFP construct contains unc-119 cDNA) and N2 (together with rol-6 marker). We have ruled out the possibility that the lack of positive animals in the F1 generation is due to faulty microinjections because parallel microinjections done only with a

rol-6 marker resulted in several dozen F1 animals with the roller phenotype. Therefore, we concluded that the provided DNA was probably toxic to the worms, most likely due to the *myo-3* promoter driving robust expression of *vit-2*, the accumulation of which can negatively affect worm muscle development.

12. I am not so sure that developing embryos per se induce muscular exophoresis (i.e., do they need to be 'developing'? Or is presence enough?).

Our observations show that the mere presence of embryos is not sufficient for the induction of exophoresis. Under conditions where fertilized, but arrested embryos appeared in utero, e.g., after FUDR treatment or RNAi depletion of the UNC-45 myosin co-chaperone, exophoresis is inhibited. Therefore, we think that developing embryos are directly or indirectly responsible for the induction of exophoresis.

13. The data in Fig. 3 A-C is consistent with the cells of the uterus inducing the exophoresis, perhaps by detecting the presence of the eggs through stretch gated ion channels.

*To verify whether the accumulation of embryos and thus uterus stretching would be sufficient to induce exophoresis, we performed the following experiment. Worms were grown on *egl-1* RNAi feeding plate, whose depletion induces egg retention in the uterus. Under control conditions, this enhances the production of exophers. However, when EGL-1 depleted worms were treated with FUDR, there was no exophoresis, despite the accumulation of embryos in the uterus, and thus potential induction of stretch gated ion channels (Figure EV2B-C). This result indicates that the appearance of developing embryos in the uterus, rather than its stretching, triggers the formation of muscle exophers.*

14. But, then the authors show embryonic EXTRACT can induce exophoresis! This is very surprising, and based on a result (Fig. 3E) that is not, statistically, very strong. The experiments were not done blind, which adds to the concern of possible unintentional bias. I would like to see this experiment repeated in a double-blind fashion and a potential mechanism for how something in egg lysate communicates to the muscle cells proposed. Are extracts able to induce exophoresis in males? If so, you might have a nice assay to identify this mystery factor. If not, I would revisit the possibility that egg sensing by the uterine cells underlies the effect seen. The authors state that 'molecules that diffuse from embryos' induce exophoresis. In addition to the comments above, the experiment (adding sonicated embryo lysate) does not rule out a role for eggshell components.

We verified whether, in the presence of the embryo extract, males would form exophers. However, this condition was not sufficient (Figure EV2A). We believe that exophoresis is regulated at two levels. First, oocyte fertilization and the subsequent developing embryo in the uterus is crucial for the activation of exophoresis. Second, exophoresis is achieved through embryo-derived metabolite sensing coupled to feedback regulation of exophoresis intensity. This conclusion is supported by the observation that exopher formation can be enhanced by exposure of the adult hermaphrodite to the embryo lysate (Figure EV2A) or embryo-

preconditioned solution (Figure 3E). In addition, the accumulation of actively dividing embryos in utero, and thus the compounds they potentially release, enhances exopher production. This also occurs upon increased permeability of the embryo permeability layer via emb-27 RNAi knockdown (Figure 3F), which likely contributed to the more efficient secretion of metabolites from embryos that might influence exopheresis activity. In this way, the eggshell component is depleted, thus the eggshell does affect exopher production, but indirectly, as a regulator of embryo permeability. All of these experiments were repeated in a double-blind fashion.

15. Here and elsewhere, it would be better to reframe the introductory statements in neutral way, rather than looking for a preferred conclusion. For example, the sentence "Finally, we decided to confirm that even in the face of disturbed proteostasis specifically in the BWM, a signal associated with the developing embryos in the uterus would be the primary regulator of muscular exopheresis" would be better rephrased as: "Finally, we explored the role of BWM proteostasis in exopher formation".

We agree with this point. We have now rephrased such statements.

16. I question the interpretation of the experiment shown in Fig. 3F. The authors show depletion of UNC-45 by RNAi decreases exopher numbers. This could easily be due to disruption of an acto-myosin based exopher extrusion mechanism in the muscle cells. Therefore, this data does not show the 'predominant role of maturing eggs in muscular exopheresis'. (minor: delete 'maturing', replace 'eggs' with 'embryos').

To verify the role of the actomyosin network in exopheresis, we depleted UNC-54, which encodes a myosin heavy chain class II (MHC-B) that is expressed in various muscle cells, including body wall and intestinal muscles. However, UNC-54 depletion did not affect exopheresis activity, suggesting any or minimal effect of disrupted body wall muscle actomyosin filaments on this process (Figure EV2D). The myosin chaperone UNC-45 is crucial for UNC-54 folding and thick filament assembly in body wall muscle and promotes non-muscle myosin II (NMY-2) during embryonic polarity establishment. Depletion of or mutations in UNC-45 result in decreased MHC-B levels (Barral et al., 1999; Melkani et al., 2011). Since the direct depletion of MHC-B (UNC-54 RNAi) does not affect exopher generation, disorders in embryonic contractility and polarity establishment due to UNC-45 RNAi are responsible for exopheresis inhibition. Therefore, this, along with our other data, point to a predominant role of in utero developing embryo in exophers regulation.

17. The authors claim that the autophagy machinery is involved in exopher production but do not perform experiments directly addressing this claim. Additional experiments that explore the role of autophagy in exopher production would strengthen the paper.

We included new data showing that depletion of two autophagy genes, atg-7 and lgg-1, with RNAi significantly reduces the number of generated exophers indicating autophagy machinery involvement in exopheresis (Figure 1G).

Reviewer: 2

Comments to the Authors

In this study entitled "Nutritional status and fecundity are synchronised by muscular exopheresis", Turek et al. show that muscle cells expel exophers, large extracellular vesicles, to provide energy resources for the next generation. Yolk protein is secreted within exophers, which nourishes the embryos in utero and even later outside the mother. Such a pinch-off of a large part of the cytosol has been first observed by Melentijevic et al. in *C. elegans* neurons, which has been attributed to a proteostasis pathway to remove damaged and dysfunctional organelles and macromolecules from the cell. In contrast, exopheresis in muscle cells seems to have less of a stress relief purpose than a growth and reproductive purpose. This is a fascinating discovery that is of interest to a broad readership. Having said this, there are still a few points that need to be addressed:

Major points:

1. The authors need to give the exact absolute numbers of worms with exophers and exophers per worm rather than the "fold change" of exophers. Absolute numbers are much more informative, and since this is the first time this phenomenon has been characterized, it is important that the reader can estimate the frequency of these events.

We have now included the absolute numbers of exophers in the tested animals in all graphs. Additionally, we have prepared a summary table (Appendix Table S1) in which we provide the numbers of worms in experiments in which we observe exophers.

2. Some exophers are very similar to coelomocytes (e.g., Fig. 1F right panel with white arrow) and therefore it would be very helpful if the authors could show how they compare to each other (size, location, content?) and how the authors distinguish exophers from coelomocytes in their analysis.

*As for the exopher in Figure 1F, its shape and organization might be reminiscent of a coelomocyte. To visualize coelomocytes and exophers simultaneously, we introduced a fluorescent marker of the former (*unc-122::GFP*) into the exopher reporter line. This allowed us to unambiguously distinguish coelomocytes from exophers, and we used the example image in Figure 1F.*

3. The numbers of exophers or worms with exophers seem to be surprisingly variable between experiments. For example: Fig. 2A: here, the data indicate that on day 2 and 3 of adulthood, there are approx. 6-9 exophers per worm, with little variation. In contrast, the graph in Fig. 2D indicates that 30% of worms have more than 10 exophers. If this were the case, then the error bars in Fig. 2A would be much larger. Is there an explanation for this discrepancy?

In contrast to counting exophers using a stereo microscope, we are able to see more of them using a confocal microscope. In the measurements shown in the previous Figure 2D we used a confocal microscope, hence the higher value on the graph. We have now repeated the experiment using a stereo microscope for counting exophers and replaced Figure 2D with the new data, which show the same effect of FUdR on exopher formation as our previous data.

4. Fig. 2: The authors claim that exopher secretion is sex specific. However, males do not produce eggs and since these trigger exopher release, the trigger could simply be missing. Therefore, the authors should test whether embryo lysate is able to trigger exophoresis in males.

*We verified whether, in the presence of the embryo extract, males would form exophers. However, this condition was not sufficient (Figure EV2A). We believe that exophoresis is regulated at two levels. First, the fertilization of the oocyte and the subsequent developing embryo in the uterus is crucial for the activation of exophoresis. Second, that exophoresis is achieved through embryo-derived metabolite sensing coupled to feedback regulation of exophoresis intensity. This conclusion is supported by the observation that exopher formation can be enhanced by exposure of the adult hermaphrodite to embryo lysate (Figure EV2A) or embryo-preconditioned solution (Figure 3E). In addition, the accumulation of actively dividing embryos in utero, and thus the compounds they potentially release, enhances exopher production. This also occurs upon increased permeability of the embryo permeability layer via *emb-27* RNAi knockdown, which likely contributed to the more efficient secretion of metabolites from embryos that might influence exophoresis activity. In this way, the eggshell component is depleted, thus the eggshell does affect exopher production, but indirectly, as a regulator of embryo permeability. All of these experiments were repeated in a double-blind fashion.*

5. Fig. 3A: according to the literature (and our own observations) hermaphrodites have usually 10-15 eggs in the uterus. The graph shows numbers of 25 up to 50, and I cannot even imagine what a worm with this high number of eggs would look like...

*We agree that young hermaphrodites typically have 10-15 eggs in the uterus, but older animals or those with egg-laying difficulties might retain many more eggs. As described in the Materials and Methods section, in this experiment, for each worm, we first counted the muscle exophers, and in the next step, the hermaphrodite mother was placed in 10 ul drops of 1.8% hypochlorite solution. After being in the hypochlorite solution for a few minutes, when the adult animal was mostly dissolved, we counted the remaining eggs that had previously been in the animal's uterus. Since the eggshell is much more resistant to bleach than the adult worm tissues, we are confident that with this approach we are able to accurately count all of the eggs present in the uterus, and we are confident that our numbers are correct. Animals with large numbers of eggs in the uterus showed a distinct *egl* phenotype.*

Also the description "animals were 1-3 days old" is much too imprecise because the age in this range drastically influences the number of exophers (see Fig. 2A).

We do not think that the age of the worm in the range of 1 to 3 days of adulthood affects the number of exophers, but instead the number of eggs present in the uterus. In-utero egg abundance increases during the first few days of worm adulthood. Thus, we purposely selected adults between days 1 to 3 to obtain as wide a range of the number of eggs present in the uterus as possible (starting at 0 for young adult worms that were just after their last molt). In addition, from this population, we also specifically selected individuals with a very high number of eggs in the uterus (as pointed out by the reviewer) to further increase the area range. This approach allowed us to calculate the Pearson correlation coefficient from a more diverse data set. Therefore, we decided to keep the data shown in Figure 3A as is.

6. Fig. 3F: There are too many variables in this experiment, which does not allow any conclusion at the moment, since *unc-45* RNAi not only reduces proteostasis but also affects the development of the embryos. The authors need to further dissect this and treat animals with *unc-45* RNAi in the presence of egg lysates. This way, they could separate the effect on muscle proteostasis from a potential negative growth signal of the affected embryos.

We performed the experiment suggested by the reviewer, but did not observe a significant change in the number of exophers after incubating *unc-45* knocked down animals in embryos lysate.

Figure 1. Incubation of animals treated with *unc-45* RNAi in embryo lysate does not affect the exophers level. $n = 25$ and 31 animals; two biological replicates. Data are shown as mean \pm SEM. ns – not significant; Mann-Whitney test.

However, we have performed additional control for the experiment with worms treated with *unc-45* RNAi. The myosin chaperone *UNC-45* is crucial for *UNC-54* folding and thick filament assembly in body wall muscle and promotes non-muscle myosin II (*NMY-2*) during embryonic polarity establishment. Depletion or mutations in *UNC-45* result in decreased MHC-B levels

(Barral et al., 1999; Melkani et al., 2011). To verify the role of the actomyosin network in exophoresis, we depleted UNC-54, which encodes a myosin heavy chain class II (MHC-B) that is expressed in many muscle cell classes, including body wall and intestinal muscles. However, UNC-54 depletion did not affect exophoresis activity, suggesting any or minimal effect of disrupted body wall muscle actomyosin filaments on this process (Figure EV2D). Since the direct depletion of MHC-B (UNC-54 RNAi) does not affect exopher generation, disorders in embryonic contractility and polarity establishment due to UNC-45 RNAi are responsible for exophoresis inhibition. Therefore, these observations and our other data point to a predominant role of in utero developing embryos in exopher regulation. We have decided to include abovementioned results in the manuscript and present the data of unc-45 RNAi and embryo lysate only in the letter.

7. Fig. 4H: It is surprising that the VIT-2::GFP is highly abundant in embryos, but no RPN-5::RFP is detected in the embryos. If the entire content of the exophers were taken up by embryos one would expect to detect both, RFP and GFP staining. Is there a selective uptake of exopher content? How could that happen? Alternatively, one could imagine that the transfer of VIT-2 is mediated by another parallel pathway, distinct from exophers...

RME-2 is a specific yolk receptor in oocytes, and depletion of RME-2 leads to inhibition of yolk uptake by oocytes and a dramatic reduction in exophoresis. Thus, we think that muscle-produced VIT-2 is supplied to oocytes via the RME-2-dependent pathway. We wanted to visualize the delivery of the muscle vitellogenin to the oocytes. We were very keen on this result and therefore spent the last months generating a worm line in which fluorescently labelled VIT-2 would be expressed from the myo-3 promoter. This would allow us to follow the trafficking of muscle-derived vitellogenin, hoping to visualize it in oocytes. We generated large (11 kb) Pmyo-3::vit-2::GFP DNA constructs that we intended to introduce into worms in two ways. Single copy insertion via CRISPR/Cas-9-based SKI LODGE system (Silva-Garcia et al., 2019) and microinjection with the created construct. Unfortunately, our several rounds of microinjections with vit-2::GFP template DNA for SKI LODGE system, injected together with rol-6 marker, did not lead to any animals with the roller phenotype. Moreover, microinjections of the Pmyo-3::vit-2::GFP DNA construct also did not result in any animal expressing vit-2::GFP. We injected it into two different worm strains, unc-119(ed3) mutants (Pmyo-3::vit-2::GFP construct contains unc-119 cDNA) and N2 (together with rol-6 marker). We have ruled out the possibility that the lack of positive animals in the F1 generation is due to faulty microinjections because parallel microinjections done only with a rol-6 marker resulted in several dozen F1 animals with the roller phenotype. Therefore, we concluded that the provided DNA was probably toxic to the worms, most likely due to the myo-3 promoter driving robust expression of vit-2, the accumulation of which can negatively affect worm muscle development. Deletion of the phagocytic receptor CED-1 does not affect the number of exophers (Figure EV4A), but we noted that it leads to an increase in the amount of VIT-2::GFP in embryos (Figure 4H). This might suggest that scavenger cells can capture exophers, but this is not the main distribution pathway of exopher cargo. The lack of a visible RPN-5::RFP signal in the embryo is probably due to its inability to perform endocytosis in the oocyte as there is no corresponding/specific receptor.

Minor points:

1. Fig. 1 legend (I): The description "proteostasis induced via oxidative stress [...]" is not correct. Please rephrase into "Challenging proteostasis via [...]".
2. Fig. 2B: labeling is wrong (+/- offspring).
3. Fig. S2: The title misleading and should rather read "Active exopheresis does NOT affect worm motility".
4. The authors misspell "exopheresis" with "exophoresis" several times throughout the text (e.g. in the title of Fig. S2, see above). Please correct.
5. Line 162: "exopher" should be plural.

We have made appropriate changes in the text.

Reviewer: 3

Comments to the Authors

Turek, Piechota et al. demonstrate that body wall muscle releases 2-15 μm cell fragments called exophers containing proteasomes, yolk proteins, and damaged mitochondria. Similar structures were previously reported to export waste for neurons and cardiac muscle, but the authors provide evidence that there are different subpopulations of exophers and that many body wall muscle exophers have trophic functions. They find that exopher production is regulated by female fertility or embryo production, while embryos seem to benefit nutritionally from increased exopher production. Given the unpublished report of exophers released from skeletal muscle in mice, these findings in *C. elegans* are likely to be conserved in mammals and provide important new insight into organismal physiology.

Major points

1. Is the right panel of Fig. 1F showing a coelomocyte? It looks like the marker has been endocytosed by a round cell and is found in many endosomes. It does not look consistent with exopher fragmentation as the authors describe in the text. Also, if the two panels are different time points, the intervening time should be labeled on the figure or given in the figure legend.

*As for the exopher in Figure 1F, its shape and organization might be reminiscent of a coelomocyte. To visualize coelomocytes and exophers simultaneously, we introduced a fluorescent marker of the former (*unc-122::GFP*) into the exopher reporter line. This allowed us to unambiguously distinguish coelomocytes from exophers, and we used the example image in Figure 1F.*

2. Fig 1H-I, 2B, 2D, 3B-C, 3E-F, S1B, S3A - Data should be presented as absolute numbers of exophers per animal like Fig. S1A, not fold change. This is especially relevant to see how exopher number in the various classes of *fem-1* mutants compare to control animals.

Also, it is important to consider that a change in exopher number can be a result of a change in exopher production, exopher clearance, or both. Therefore it is important to report data throughout the manuscript in terms of exopher number, not exopher production.

We have now included the absolute numbers of exophers in the tested animals in all graphs. Additionally, we have prepared a summary table (Appendix Table S1) in which we provide the numbers of worms in experiments in which we did not observe exophers.

3. Fig. 3F is missing a control that exopher production by muscle cells is not altered by paralysis. For example, *unc-54* RNAi could be used as a control targeting myosin that does not disrupt embryonic development.

The myosin chaperone UNC-45 is crucial for UNC-54 folding and thick filament assembly in body wall muscle and promotes non-muscle myosin II (NMY-2) during embryonic polarity establishment. Depletion or mutations in UNC-45 result in decreased MHC-B levels (Barral et al., 1999; Melkani et al., 2011). To verify the role of the actomyosin network in exophoresis, we depleted UNC-54, which encodes a myosin heavy chain class II (MHC-B) that is expressed in many muscle cell classes, including body wall and intestinal muscles. However, UNC-54 depletion did not affect exophoresis activity, suggesting any or minimal effect of disrupted body wall muscle actomyosin filaments on this process (Figure EV2D). Since the direct depletion of MHC-B (UNC-54 RNAi) does not affect exopher generation, disorders in embryonic contractility and polarity establishment due to UNC-45 RNAi are responsible for exophoresis inhibition. Therefore, these observations and our other data point to a predominant role of in utero developing embryos in exopher regulation.

4. How do the authors propose that vitellogenins in 2-15 μ m exophers are being fed to oocytes if not through exopher phagocytosis. Would a *ced-1* mutant strain still show the increase in VIT-2::GFP in high exopher worms?

RME-2 is a specific yolk receptor in oocytes, and depletion of RME-2 leads to inhibition of yolk uptake by oocytes and a dramatic reduction in exophoresis. Thus, we think that muscle-produced vitellogenins are supplied to oocytes via the RME-2-dependent pathway. Deletion of the phagocytic receptor CED-1 does not affect the number of exophers (Figure EV4A), but we noted that it leads to an increase in the amount of VIT-2::GFP in embryos (Figure 4H). This might suggest that scavenger cells can capture exophers, but this is not the main distribution pathway of exophers cargo.

Minor points

1. Fig. S1A - Would be better represented as paired data showing the number of proteasome-containing exophers and the corresponding number of mitochondria-containing exophers in each animal.

We have made appropriate changes in the figure.

2. 73 - "probably not compromised mitochondria"

Does this mean the mitochondria had normal appearing cristae? In other exopher studies, they tested mitochondrial function by looking at oxidation or other functional assays. Either reword to describe the assay or test mitochondrial function.

To avoid confusion, we have made appropriate changes in the text.

3. Fig. 2B - Offspring label is inconsistent with the text.

We have made appropriate changes in the figure.

4. Line 115-118 - The text is not clear how the authors were scoring exophers vs "exopher-like" vs "non-extruded exophers". What is the distinction in Fig. 2E? Are these images projections or a single focus in the plane of the muscle?

We have described in the main text and legend to Figure 2E, that exopher marker-labeled structures that resemble exophers (size/shape) were not excreted by the BWM of FUDR-treated hermaphrodites and males. This potentially indicates an active process of segregation and storage of material that constitutes the natural contents of exophers, but not an active mechanism for their excretion outside the muscle. Thus, we have suggested that males are 'devoid of mechanisms' that trigger exopher expulsion. However, we understand that we might have too hastily characterized these intramuscular objects as non-secreted exophers. We have now rephrased the statement to "We also found that hermaphrodites treated with FUDR often contained structures in their BWM that appeared like segregated exopher cargo (Figure 2E, middle and right panels). Interestingly, we detected similar objects in males (Figure 2E, left panel) that, as in sterile hermaphrodites, are not excreted by the BWM."

All images are projections from a Z-stack.

5. Fig. 3A - 2-3 day old adults hold more embryos in the uterus than young adults and the authors showed in Fig. 2A that exophers peak at day 2-3 of adulthood. This analysis should not include data from day 1 adults to avoid conflating the effect of holding eggs with developmental timing.

*We agree that young adults hold fewer eggs in the uterus than 2- to 3-day-old animals. However, in our experiment, we wanted to cover the wide range for the number of eggs present in utero (starting from 0 for young adult worms, which were just after the last molt). Therefore, we have deliberately selected animals that were 1 to 3 days old. Moreover, we have also specifically picked animals with very high numbers of eggs in the uterus to even further increase the plot range from this population. This approach allowed us to calculate the Pearson correlation coefficient from a more diverse set of data. To avoid conflating the effect of egg retention in the uterus with developmental timing, we have manipulated their presence in the uterus by knocking down *goa-1*, *egl-1*, and *egl-4* (Figure 3B). These animals were of the same age (2 days old); however, they showed opposite phenotypes regarding egg retention: *goa-1* knockdown worms held fewer eggs in the uterus than wild-type worms, while *egl-1* and*

egl-4 knockdown worms retained more eggs in the uterus than wild type worms. Therefore, we decided to keep the data presented in Figure 3A as is.

6. Could the authors briefly discuss what is known about the effect of egg-laying mutants and starvation on oocyte production, maturation, or ovulation? The observed effects on exophers could be tertiary to secondary effects on germ cell biology. Do strains with increased germ cell divisions that make germ line tumors, such as *glp-1(gof)*, have altered exopher numbers?

[Unpublished data and its description removed at the author's request]

We thank the reviewer for the suggestion regarding germline tumors, we are currently conducting further studies related to this phenomenon. Because of that reason, we decided not to include this data in our manuscript but instead we present them in the letter. On the other hand, just 6 hours of starvation is sufficient to induce germ cell apoptosis in one-day-old adult worms and causes egg retention and bagging (Lascarez-Lagunas et al., 2014). In this condition, we observed a significant increase in the number of exophers, likely related to embryo retention in utero, which might lead to increased concentrations of embryo-produced compounds that might regulate exopheresis activity.

7. Lysed embryos in Fig. 3D-E are not an approximation for molecules diffusing from developing embryos. Are the authors proposing a small molecule signal that is able to cross the permeability barrier of the eggshell? Conditioned media incubated with embryos would be more comparable. Either repeat experiment or rephrase. It may also be worth considering the

mechanical impact on the body wall muscle of holding embryos in the uterus in *egl* mutants, etc.

*As suggested by the reviewer, we have repeated the experiment with a conditioned media incubated with embryos (Figure 3E). We have obtained the same result as for the incubation with lysed embryos (Figure EV2A). Moreover, upon increased permeability of the embryo permeability layer via *emb-27* RNAi depletion, which likely contributed to more efficient secretion of metabolites/small molecules from embryos, we could observe increased exopheresis activity.*

*To verify whether the accumulation of embryos and thus uterus stretching (mechanical impact) would be sufficient to induce exopheresis, we performed the following experiment. Worms were grown on *egl-1* RNAi feeding plate, as *egl-1* depletion induces egg retention in the uterus. Under control conditions, this enhances exopher production. However, when *egl-1*-depleted worms were treated with FUdR, there was no induction of exopheresis, despite the accumulation of embryos in the uterus (mechanical impact), and thus potential induction of stretch gated ion channels (Figure EV2B-C). This result indicates that the appearance of developing embryos in the uterus, rather than mechanical impact, triggers the formation of muscle exophers.*

8. Line 197- The authors show no data for autophagy machinery. Missing reference(s)?

*We now included data showing that RNAi depletion of two autophagy genes, *atg-7* and *lgg-1*, significantly reduces the number of generated exophers, indicating autophagy machinery involvement in exopheresis (Figure 1G).*

9. Line 545 - How do the HT115 bacteria without a vector grow on carbenicillin plates?

We gave an inconsistent description. RNAi control experiments were performed with HT115 bacteria containing empty plasmid, hence the information regarding carbenicillin in the plates. We have made appropriate changes in the text.

We thank Reviewers for insightful comments that improved our manuscript and for supporting its publication in EMBO Reports.

With best regards,

Michał Turek and Wojciech Pokrzywa

Dear Wojtek,

I am happy to tell you that we have now received the full set of referee reports on your revised manuscript (copied below).

As you will see, all referees are very positive about the study and support publication after clarifying some issues in figures and toning down/adjusting the conclusions regarding the presence of developing embryos as trigger of exopheresis.

Browsing through the manuscript myself, I also noticed a few editorial things that we need before we can proceed with the official acceptance of your study.

- Please remove the figures from the manuscript file.
- Please provide up to 5 keywords
- Please note that a Data Availability section at the end of Materials and Methods is now mandatory. In case you have no data that requires deposition in a public database, please state so instead of refereeing to the database.
See also < <https://www.embopress.org/page/journal/14693178/authorguide#dataavailability>>). Please note that the Data Availability Section is restricted to new primary data that are part of this study.
- Please add a conflict of interest paragraph
- Please include the information on funding in the Acknowledgement section.
- Appendix: please combine the two Appendix tables and their legends into one pdf called "Appendix". This pdf needs a title page with a table of content including page numbers. Please change the name of the tables to Appendix Table S1 and S2 and also correct the respective callouts in the text.
- Movie: please rename it to "Movie EV1". Please provide a legend for the movie as simple README.txt file, zip the movie with its legend and then upload the zipped file.
- The "Methods" heading should be corrected to Materials and Methods.
- I notice one citation of a manuscript that has been posted to bioRxiv (Hualin F et al, 2019). Please cite it the following way:
 - Authors, (YEAR) article title. bioRxiv doi: xxx/xxx [PREPRINT]
- Our production/data editors have asked you to clarify several points in the figure legends (see attached document). Please incorporate these changes in the attached word document and return the revised file with tracked changes with your final manuscript submission. I have also taken the liberty to suggest a different title. Could you please review it?
- We note that the statistical comparison of results shown in Figures 1I, 3B, 4E and EV2A/D is based on n = 2 biological replicates. Please note that our editorial policies do not recommend the

use of statistics if $n < 3$. Therefore, please either remove the statistical comparison from these figure panels or provide additional replicates. This also applies to Fig. EV4B, in case these data are also from $n = 2$.

- During our routine image integrity analysis we noticed that the graphs shown in Fig. 4B and EV3 are very similar to each other. To avoid any ambiguity, I kindly ask you to provide the source data used to generate these graphs.

- Finally, EMBO reports papers are accompanied online by A) a short (1-2 sentences) summary of the findings and their significance, B) 2-3 bullet points highlighting key results and C) a synopsis image that is 550x200-600 pixels large (width x height) in .png format. You can either show a model or key data in the synopsis image. Please note that the size is rather small and that text needs to be readable at the final size. Please send us this information along with the revised manuscript.

With kind regards,

Martina

Referee #1:

The authors have addressed my concerns. The revisions provide much better substantiation for these intriguing results.

Referee #2:

Most of my previous concerns have been sufficiently addressed, but there are still two points I would like to emphasize.

Regarding point 2: Comparing Figs. 1A and 1D: the left structure (near the muscle cell) in Fig. 1A looks much more like the green-stained coelomocyte in Fig. 1D than the red exopher in Fig. 1D, which clearly lacks this characteristic pronounced "vesicle signature" inside. Choosing an alternative Fig. 1A with exophores that resemble the exopher rather than the coelomocyte in Fig. 1D would eliminate potential concerns of misidentification.

Regarding point 4: You state in your response that "fertilization of the oocyte and subsequent development of the embryo in utero is critical for activation of exopheresis." You make this strong statement repeatedly throughout the text. However, in your response to point 6 of reviewer 3, you now provide data in Fig. 2 for reviewers only, to which you note "Interestingly, exopheresis was active in *glp-1(ar202)* worms at the restrictive temperature despite the absence of embryos in utero." Given these data, it is surprising that you continue to make the claim that fertilized embryos would be indispensable. It is fine if you choose to not include these data here and save

them for another study, but you cannot simply ignore these data. The strong wording about the essential role of fertilized embryos in utero regarding exopher production needs to be toned down throughout the text because this is misleading in light of these new findings. Fertilized embryos in utero are clearly an important trigger and regulator of exopheresis, but apparently neither essential nor crucial.

Referee #3:

Turek et al. demonstrate that body wall muscle releases 2-15 μm cell fragments called exophers containing proteasomes, yolk proteins, and damaged mitochondria. Similar structures were previously reported to export waste for neurons and cardiac muscle, but the authors provide evidence that body wall muscle exophers may have trophic functions. They find that exopher production is regulated by female fertility or embryo production and provide evidence that embryos benefit from increased exopher production. Given the unpublished report of exophers released from skeletal muscle in mice, these findings in *C. elegans* are likely to be conserved in mammals and provide important new insight into organismal physiology.

The authors have addressed all concerns from the initial round of review and significantly improved their manuscript. It is an exciting addition to a new field.

Dr. Martina Rembold
Senior Editor
EMBO reports

Wojciech Pokrzywa, PhD
Head of the Laboratory of Protein Metabolism
International Institute of Molecular and Cell
Biology
4 Ks. Trojdena Street
02-109 Warsaw
Poland
wpokrzywa@iimcb.gov.pl
+48225970743

Dear Dr. Martina Rembold

Please find a detailed description of the edited paragraphs below (the reviewers' comments are **bold** and our responses are *italic*):

Referee #1:

The authors have addressed my concerns. The revisions provide much better substantiation for these intriguing results.

Referee #2:

Most of my previous concerns have been sufficiently addressed, but there are still two points I would like to emphasize.

Regarding point 2: Comparing Figs. 1A and 1D: the left structure (near the muscle cell) in Fig. 1A looks much more like the green-stained coelomocyte in Fig. 1D than the red exopher in Fig.1D, which clearly lacks this characteristic pronounced "vesicle signature" inside. Choosing an alternative Fig. 1A with exophores that resemble the exopher rather than the coelomocyte in Fig. 1D would eliminate potential concerns of misidentification.

As suggested, we have substituted the image in Fig. 1A.

Regarding point 4: You state in your response that "fertilization of the oocyte and subsequent development of the embryo in utero is critical for activation of exopheresis." You make this strong statement repeatedly throughout the text. However, in your response to point 6 of reviewer 3, you now provide data in Fig. 2 for reviewers only, to which you note "Interestingly, exopheresis was active in *glp-1(ar202)* worms at the restrictive temperature despite the absence of embryos in utero." Given these data, it is surprising that you continue to make the claim that fertilized embryos would be indispensable. It is fine if you choose to not include these data here and save them for another study, but you cannot simply ignore these data. The strong wording about the essential role of fertilized embryos in utero regarding exopher production needs to be toned down throughout the text because this is misleading in

light of these new findings. Fertilized embryos in utero are clearly an important trigger and regulator of exopheresis, but apparently neither essential nor crucial.

In the revised manuscript, the role of embryos in exopher production was already toned down throughout the text. We did not use terms suggesting their critical role but their participation in the regulation of exopheresis. The sentence "fertilization of the oocyte and subsequent development of the embryo in utero is critical for activation of exopheresis." was used only in the rebuttal letter. We are sorry for the confusion.

Referee #3:

Turek et al. demonstrate that body wall muscle releases 2-15 μm cell fragments called exophers containing proteasomes, yolk proteins, and damaged mitochondria. Similar structures were previously reported to export waste for neurons and cardiac muscle, but the authors provide evidence that body wall muscle exophers may have trophic functions. They find that exopher production is regulated by female fertility or embryo production and provide evidence that embryos benefit from increased exopher production. Given the unpublished report of exophers released from skeletal muscle in mice, these findings in *C. elegans* are likely to be conserved in mammals and provide important new insight into organismal physiology.

The authors have addressed all concerns from the initial round of review and significantly improved their manuscript. It is an exciting addition to a new field.

Below we are providing a response to the Editor comments:

- Please remove the figures from the manuscript file.

Done.

- Please provide up to 5 keywords

C. elegans, muscle, exophers, embryo, development

- Please note that a Data Availability section at the end of Materials and Methods is now mandatory. In case you have no data that requires deposition in a public database, please state so instead of referring to the database.

See also

< <https://www.embopress.org/page/journal/14693178/authorguide#dataavailability>>). Please note that the Data Availability Section is restricted to new primary data that are part of this study.

Done.

- Please add a conflict of interest paragraph

Done.

- Please include the information on funding in the Acknowledgement section.

Done.

- Appendix: please combine the two Appendix tables and their legends into one pdf called "Appendix". This pdf needs a title page with a table of content including page numbers. Please change the name of the tables to Appendix Table S1 and S2 and also correct the respective callouts in the text.

Done.

- Movie: please rename it to "Movie EV1". Please provide a legend for the movie as simple README.txt file, zip the movie with its legend and then upload the zipped file.

Done.

- The "Methods" heading should be corrected to Materials and Methods.

Done.

- I notice one citation of a manuscript that has been posted to bioRxiv (Hualin F et al, 2019). Please cite it the following way:

- Authors, (YEAR) article title. bioRxiv doi: xxx/xxx [PREPRINT]

Done.

- Our production/data editors have asked you to clarify several points in the figure legends (see attached document). Please incorporate these changes in the attached word document and return the revised file with tracked changes with your final manuscript submission. I have also taken the liberty to suggest a different title. Could you please review it?

We have incorporated the required changes.

- We note that the statistical comparison of results shown in Figures 1I, 3B, 4E and EV2A/D is based on $n = 2$ biological replicates. Please note that our editorial policies do not recommend the use of statistics if $n < 3$. Therefore, please either remove the statistical comparison from these figure panels or provide additional replicates. This also applies to Fig. EV4B, in case these data are also from $n = 2$.

The N and n numbers and statistical analysis process is highlighted in the legend and the corresponding section in materials and methods.

- During our routine image integrity analysis, we noticed that the graphs shown in Fig. 4B and EV3 are very similar to each other. To avoid any ambiguity, I kindly ask you to provide the source data used to generate these graphs.

Mobility analysis was performed using the WormLab system, which automatically measures many of the worms' movement parameters. After averaging, data from WormLab software was transferred to GraphPad software to perform statistical analysis. We included the original GraphPad file in the system. Returning to the results, there are subtle differences between track length and center point speed, but, intuitively, the distribution will look similar, as the movement speed per unit time also corresponds to the distance covered.

- Finally, EMBO reports papers are accompanied online by A) a short (1-2 sentences) summary of the findings and their significance, B) 2-3 bullet points highlighting key results and C) a synopsis image that is 550x200-600 pixels large (width x height) in .png format. You can either show a model or key data in the synopsis image. Please note that the size is rather small and that text needs to be readable at the final size. Please send us this information along with the revised manuscript.

Done.

With best regards,

Michał Turek and Wojciech Pokrzywa

Dr. Wojciech Pokrzywa
International Institute of Molecular and Cell Biology in Warsaw
Laboratory of Protein Metabolism
Warsaw
Poland

Dear Wojtek,

Thank you for sending the revised files and for making further corrections. I am now very pleased to accept your manuscript for publication in the next available issue of EMBO reports. Thank you for your contribution to our journal.

At the end of this email I include important information about how to proceed. Please ensure that you take the time to read the information and complete and return the necessary forms to allow us to publish your manuscript as quickly as possible.

As part of the EMBO publication's Transparent Editorial Process, EMBO reports publishes online a Review Process File to accompany accepted manuscripts. As you are aware, this File will be published in conjunction with your paper and will include the referee reports, your point-by-point response and all pertinent correspondence relating to the manuscript.

If you do NOT want this File to be published, please inform the editorial office within 2 days, if you have not done so already, otherwise the File will be published by default [contact: emboreports@embo.org]. If you do opt out, the Review Process File link will point to the following statement: "No Review Process File is available with this article, as the authors have chosen not to make the review process public in this case."

Should you be planning a Press Release on your article, please get in contact with emboreports@wiley.com as early as possible, in order to coordinate publication and release dates.

Thank you again for your contribution to EMBO reports and congratulations on a successful publication. Please consider us again in the future for your most exciting work.

Kind regards,
Martina

THINGS TO DO NOW:

You will receive proofs by e-mail approximately 2-3 weeks after all relevant files have been sent to

our Production Office; you should return your corrections within 2 days of receiving the proofs.

Please inform us if there is likely to be any difficulty in reaching you at the above address at that time. Failure to meet our deadlines may result in a delay of publication, or publication without your corrections.

All further communications concerning your paper should quote reference number EMBOR-2020-52071V3 and be addressed to emboreports@wiley.com.

Should you be planning a Press Release on your article, please get in contact with emboreports@wiley.com as early as possible, in order to coordinate publication and release dates.

YOU MUST COMPLETE ALL CELLS WITH A PINK BACKGROUND ↓
PLEASE NOTE THAT THIS CHECKLIST WILL BE PUBLISHED ALONGSIDE YOUR PAPER

Corresponding Author Name: Wojciech Pokrzywa
Journal Submitted to: EMBO Reports
Manuscript Number: EMBOR-2020-52071V1